# Manifold Density Estimation via Generalized Dequantization

## Abstract

Density estimation is an important technique for characterizing distributions given observations. Much existing research on density estimation has focused on cases wherein the data lies in a Euclidean space. However, some kinds of data are not well-modeled by supposing that their underlying geometry is Euclidean. Instead, it can be useful to model such data as lying on a *manifold* with some known structure. For instance, some kinds of data may be known to lie on the surface of a sphere. We propose a method for estimating densities on manifolds which combines normalized flows in a Euclidean space, a change of variables, and marginalization. The method is applicable where the appropriate change of variables is tractable, for example the sphere, tori, and the orthogonal group.

## 1 Introduction

Certain kinds of data are not well-modeled under the assumption of an underlying Euclidean geometry. Examples include data with a fundamental directional structure, data that represents transformations of Euclidean space (such as rotations and reflections), data that has periodicity constraints or data that represents hierarchical structures. In such cases, it is important to explicitly model the data as lying on a *manifold* with a suitable structure; for instance a sphere would be appropriate for directional data, the orthogonal group for rotations and reflections, and the torus captures structural properties of periodicity.

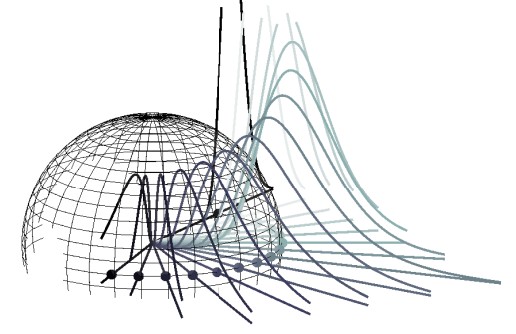

Figure 1: To generate samples from a density on the sphere $\mathbb{S}^2$, we use a normalizing flow to sample from $\mathbb{R}^3$ and project to $\mathbb{S}^2$. To compute the density at $y \in \mathbb{S}^2$, we "dequantize" from the sphere to $\mathbb{R}^3$ using an auxiliary density over $r \in \mathbb{R}_+$.

The first contribution of this work is to express density estimation on manifolds as a form of dequantization. Given a probability density in an ambient Euclidean space, one can obtain the density on the manifold by performing a *manifold change-of-variables* in which the manifold structure appears and then projecting out any auxiliary structures. This marginalization can be viewed as analogous to "quantization" where, for instance, continuous values are discarded and only rounded integer values remain. In this view the auxiliary structure defines how the manifold could be "dequantized" into the ambient Euclidean space. By marginalizing the auxiliary dimensions, the marginal distribution on the manifold is obtained. In practice, however, one has only the manifold-constrained observations from an unknown distribution on the manifold. A second contribution of this work is to formulate the density estimation as a learning problem on the ambient Euclidean space. We show how to invoke the manifold change-of-variables and perform the marginalization along the auxiliary dimensions to obtain effective estimates of the density on the manifold. An advantage of our dequantization approach is that it *allows one to utilize any expressive density directly on the ambient Euclidean space* (e.g., RealNVP (Dinh et al., 2017), neural ODEs (Chen et al., 2018; Grathwohl et al., 2018) or any other normalizing flow (Kobyzev et al., 2020)); the dequantization approach does not require a practitioner to construct densities intrinsically on the manifold. We emphasize that our theory can be applied to any embedded manifold, provided one can identify a suitable auxiliary manifold structure and

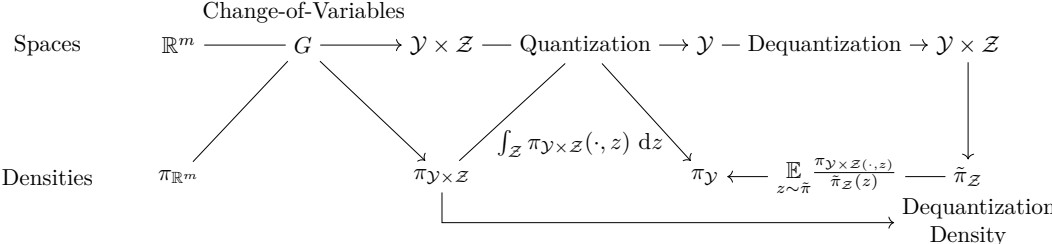

Figure 2: The dequantization roadmap. In the first row, we begin with $\mathbb{R}^m$. This Euclidean space is transformed into the product of manifolds $\mathcal{Y} \times \mathcal{Z}$ via a change-of-variables $G : \mathbb{R}^m \to \mathcal{Y} \times \mathcal{Z}$. Quantization takes the product manifold $\mathcal{Y} \times \mathcal{Z}$ to its $\mathcal{Y}$-component alone. In the second row, we begin with a probability density $\pi_{\mathbb{R}^m}$ defined on $\mathbb{R}^m$. Under the change-of-variables $G$ we obtain a new probability density $\pi_{\mathcal{Y} \times \mathcal{Z}}$ which is related to $\pi_{\mathbb{R}^m}$ by the manifold change-of-variables eq. (6). Quantizing $\mathcal{Y} \times \mathcal{Z}$ marginalizes out the $\mathcal{Z}$-component of $\pi_{\mathcal{Y} \times \mathcal{Z}}$. We similarly introduce a dequantization density $\tilde{\pi}_{\mathcal{Z}}$ and compute the marginal density on $\mathcal{Y}$ via importance sampling.

manifold change-of-variables to Euclidean space; finding a suitable auxiliary structure may be a challenge in some cases. We focus our attention on several important matrix manifolds which are common in practice and provide suitable structures for them.

## 2 Illustrative Example: Sphere

We first offer a simple example to illustrate how the method may be applied to obtain densities on the sphere. Let $\mathbb{S}^2 \subset \mathbb{R}^3$ represent the 2-dimensional sphere viewed naturally as an embedded manifold of 3-dimensional Euclidean space. Excluding the point $(0, 0, 0) \in \mathbb{R}^3$, observe that every other point $x \in \mathbb{R}^3$ may be uniquely identified with a point $s \in \mathbb{S}^2$ and positive real number $r \in \mathbb{R}_+$ such that $x = rs$. We note that this isomorphism of $\mathbb{R}^3 \setminus \{0\}$ and $\mathbb{S}^2 \times \mathbb{R}_+$ is a smooth isomorphism with smooth inverse; therefore, it is a *diffeomorphism*. Diffeomorphisms comprise the set of isomorphisms that we will primarily consider in this work. Thus, $\mathbb{S}^2 \times \mathbb{R}_+$ represents a *spherical coordinate system* for $\mathbb{R}^3 \setminus \{0\}$. If $\pi_{\mathbb{R}^3}(x)$ is a density on $\mathbb{R}^3$ (equivalent to $\mathbb{R}^3 \setminus \{0\}$ up to a set of Lebesgue measure zero), we can apply the standard change-of-variables formula in order to obtain a density on $\mathbb{S}^2 \times \mathbb{R}_+$:

$$\pi_{\mathbb{S}^2 \times \mathbb{R}_+}(s, r) = r^2 \cdot \pi_{\mathbb{R}^3}(rs), \tag{1}$$

where $r^2$ is the associated Jacobian determinant accounting for changes in volume; a derivation of this Jacobian determinant is included in appendix N. We will refer to $\pi_{\mathbb{R}^3}$ as the *ambient Euclidean* distribution.

By sampling $x \in \mathbb{R}^3$ with density $\pi_{\mathbb{R}^3}(x)$, converting to the spherical coordinate system and discarding the radius $r$, we obtain a sample from the marginal distribution $\pi_{\mathbb{S}^2}$ on the sphere $\mathbb{S}^2$. The density $\pi_{\mathbb{S}^2}$ is given by:

$$\pi_{\mathbb{S}^2}(s) = \int_{\mathbb{R}_+} \pi_{\mathbb{R}^3}(rs) r^2 \ \mathrm{d}r = \int_{\mathbb{R}_+} \pi_{\mathbb{S}^2 \times \mathbb{R}_+}(s, r) \ \mathrm{d}r \tag{2}$$

This density can be evaluated using importance sampling using a non-vanishing density $\tilde{\pi}_{\mathbb{R}_+}$ on $\mathbb{R}_+$,

$$\pi_{\mathbb{S}^2}(s) = \mathop{\mathbb{E}}_{r \sim \tilde{\pi}_{\mathbb{R}_+}} \frac{\pi_{\mathbb{S}^2 \times \mathbb{R}_+}(s, r)}{\tilde{\pi}_{\mathbb{R}_+}(r|s)}. \tag{3}$$

We note that $\tilde{\pi}_{\mathbb{R}_+}$ may depend on $s \in \mathbb{S}^2$. A visualization of these concepts is presented in fig. 1.

Equation (2) *quantizes* $\mathbb{S}^2 \times \mathbb{R}_+$ to $\mathbb{S}^2$ by integrating out the auxiliary dimension. Equation (3) describes how to marginalize $\pi_{\mathbb{S}^2 \times \mathbb{R}_+}$ over $\mathbb{R}_+$ to compute the density of a quantized point $s \in \mathbb{S}^2$; hence, we refer to $\tilde{\pi}_{\mathbb{R}_+}$ as the *dequantization density*. Taking the log of eq. (3), and invoking Jensen's inequality we can also

obtain the following lower bound on the marginal log-probability of $s$:

$$\log \pi_{\mathbb{S}^2}(s) \geq \mathop{\mathbb{E}}_{r \sim \tilde{\pi}_{\mathbb{R}_+}} \log \frac{\pi_{\mathbb{S}^2 \times \mathbb{R}_+}(s, r)}{\tilde{\pi}_{\mathbb{R}_+}(r)}. \tag{4}$$

Finally, we can use eq. (1) to express both eq. (3) and eq. (4) in terms of the density on the ambient Euclidean space $\mathbb{R}^3$.

**Learning the Distributions: Sphere Example.** In order to make these densities learnable, we introduce parameters $\theta \in \mathbb{R}^{n_{\text{amb}}}$ and $\phi \in \mathbb{R}^{n_{\text{deq}}}$ and write $\pi_{\mathbb{R}^3}(x) \equiv \pi_{\mathbb{R}^3}(x|\theta)$ and $\tilde{\pi}_{\mathbb{R}_+}(r) \equiv \tilde{\pi}_{\mathbb{R}_+}(r|\phi, s)$. For instance, $\pi_{\mathbb{R}^3}(x|\theta)$ could be a normalizing flow parameterized by $\theta$ and $\tilde{\pi}_{\mathbb{R}_+}(r|\phi, s)$ could be a log-normal distribution whose mean and variance parameters are determined by a neural network with parameters $\phi$ and input $s$.

We can use eq. (4) as an objective function for performing density estimation on the sphere. Given samples $\mathcal{D} = (s_1, \ldots, s_{n_{\text{obs}}})$ we compute a lower bound on the marginal log-probability as,

$$\mathop{\mathbb{E}}_{s \sim \text{Unif}(\mathcal{D})} \log \pi_{\mathbb{S}^2}(s) \geq \mathop{\mathbb{E}}_{s \sim \text{Unif}(\mathcal{D})} \mathop{\mathbb{E}}_{r \sim \tilde{\pi}_{\mathbb{R}_+}} \log \frac{\pi_{\mathbb{R}^3}(rs|\theta)}{\tilde{\pi}_{\mathbb{R}_+}(r|s, \phi)/r^2}. \tag{5}$$

We can then maximize the right-hand side with respect to $\theta$ and $\phi$. This yields a procedure for estimating the density on a sphere by transforming a density in an ambient Euclidean space and marginalizing over the radial dimension. The learned distribution on the ambient space $\pi_{\mathbb{R}^3}(x|\theta)$ is represented using a normalizing flow. This allows us to sample $x \sim \pi_{\mathbb{R}^3}(x|\theta)$ and to evaluate the density $\pi_{\mathbb{R}^3}(x|\theta)$. Therefore, to sample from the learned $\pi_{\mathbb{S}^2}$ we simply sample $x \sim \pi_{\mathbb{R}^3}(x|\theta)$, and then project it to the sphere: $s = x/\|x\|$.

**Evaluating the Density: Sphere Example.** Next, we may wish to evaluate the learned density $\pi_{\mathbb{S}^2}$. The normalizing flow provides us with the learned density $\pi_{\mathbb{R}^3}(x|\theta)$ in the ambient space, which immediately gives us the density $\pi_{\mathbb{S}^2 \times \mathbb{R}_+}$ through eq. (1), we must marginalize over $\mathbb{R}_+$ so as to obtain a density on $\mathbb{S}^2$. We return to eq. (2) and use the dequantization distribution $\tilde{\pi}_{\mathbb{R}_+}$ to evaluate $\pi_{\mathbb{S}^2}(s)$ using importance sampling.

## 3 Related Work

The general problem of density estimation is well studied and a full review is beyond the scope of this paper. We note early work in Breitenberger (1963); Mardia et al. (2000) on projecting the normal distribution to the circle and sphere. The most directly related work, however, is in the area of density estimation with normalizing flows and we refer readers to the review articles by Papamakarios et al. (2019) and Kobyzev et al. (2020). Recently, Horvat & Pfister (2022) used an inflation-deflation approach to modeling densities on manifolds by adding random noise.

From the perspective of estimating general densities on smooth manifolds, our work is related to Rezende et al. (2020), which considers normalizing flows on tori and spheres. Bose et al. (2020) defines a class of normalizing flows on hyperbolic spaces. Wang & Gelfand (2013) considers the density of a multivariate normal random variable projected to the sphere via the mapping $x \mapsto x/\|x\|$. For methods on connected Lie groups, one may use the exponential map in order to smoothly parameterize an element of the group by a coordinate in Euclidean space; this was the approach adopted in Falorsi et al. (2019). Lou et al. (2020) extended continuous normalizing flows to manifolds by defining and simulating ordinary differential equations on the manifold. In contrast to these approaches, the method proposed here allows for the use of any density estimation technique defined on the ambient Euclidean space. However, our procedure requires that one derive the requisite Jacobian determinants to facilitate the change of volume; we provide several examples of manifolds in which this is possible in section 5.

This work was inspired by dequantization techniques for normalizing flows. Originally introduced by Uria et al. (2013) to account for the discrete nature of pixel intensities, the basic approach added (uniform) continuous noise to discrete values. This prevented the pathological behaviour that is known to occur when fitting continuous density models to discrete data (Theis et al., 2016). The technique was extended by Ho et al. (2019) to allow the added noise distribution to be learned using a variational objective which has become critical for strong quantitative performance in image modelling. Hoogeboom et al. (2020) generalized

the variational objective used for learning the dequantization noise distribution. Recently Lippe & Gavves (2021) extended normalizing flows to categorical distributions. Our work proposes a new perspective on these approaches that the ambient Euclidean space (over which a continuous density model is learned) is a product of a discrete space (e.g., pixel intensities or discrete categories) and a continuous space which is projected (or "quantized") when data is observed. Our work is a generalization of dequantization to consider not only discrete spaces, but also other structured spaces, specifically non-Euclidean manifolds.

A previous version of this work appeared in a workshop.

## 4 Preliminaries

In order to make the discussion of the theory and algorithms more succinct, we provide in this section a brief reminder of standard notation and properties of some standard manifolds.

### 4.1 Notation

Let $\mathrm{Id}_n$ denote the $n \times n$ identity matrix. If $\mathcal{X}$ and $\mathcal{Y}$ are isomorphic sets we denote this by $\mathcal{X} \cong \mathcal{Y}$. The set of $\mathbb{R}$-valued, full-rank $n \times p$ matrices ($n \geq p$) is denoted $\mathrm{FR}(n, p)$. The set of full-rank $n \times n$ matrices is denoted $\mathrm{GL}(n)$, the generalized linear group. A matrix $\mathbf{P} \in \mathbb{R}^{n \times n}$ is positive definite if for all $x \in \mathbb{R}^n$ we have $x^\top \mathbf{P} x > 0$ for $\|x\| \neq 0$. The set of all $n \times n$ positive definite matrices is denoted $\mathrm{PD}(n)$. When $\mathbf{P} \in \mathrm{PD}(n)$ is a positive-definite matrix we denote the principal matrix square root by $\sqrt{\mathbf{P}}$.

### 4.2 Examples of Manifolds

We now consider several examples of embedded manifolds; see also appendix I for details on manifold embeddings. We focus our discussion around these particular examples because of their importance in data science applications.

#### 4.2.1 Example: Hypersphere (definitions)

The sphere in $\mathbb{R}^m$ is defined by, $\mathbb{S}^{m-1} = \left\{ x \in \mathbb{R}^m : x^\top x = 1 \right\}$. The sphere is an important manifold for data with a directional component (such as the line-of-sight of an optical receiver) or data that naturally lies on the surface of a spherical body (such as occurrences of solar flares on a star).

#### 4.2.2 Example: Torus (definition)

The torus is the product manifold of circles $\mathbb{T}^n \stackrel{\mathrm{def.}}{=} \underbrace{\mathbb{S}^1 \times \cdots \times \mathbb{S}^1}_{n \text{ times}}$. The torus $\mathbb{T}^2 = \mathbb{S}^1 \times \mathbb{S}^1$ can be embedded in $\mathbb{R}^4$ by embedding each circle individually in $\mathbb{R}^2$. The torus is an important manifold for studying systems with several angular degrees of freedom (such as applications to robotic arms) or systems with periodic boundaries (Rezende et al., 2020).

#### 4.2.3 Example: Stiefel Manifold and Orthogonal Group (definition)

The Stiefel manifold represents orthogonality constraints in a vector space. It may be regarded as the manifold of orthonormal vectors within a larger ambient Euclidean space. The Stiefel manifold can be leveraged for low-rank matrix completion. An important variant of the Stiefel manifold is the orthogonal group, which is the collection of all linear transformations of Euclidean space that preserve distance. The Stiefel manifold of order $(n, p)$ is defined by, $\mathrm{Stiefel}(n, p) \stackrel{\mathrm{def.}}{=} \left\{ \mathbf{M} \in \mathbb{R}^{n \times p} : \mathbf{M}^\top \mathbf{M} = \mathrm{Id}_n \right\}$. The $n$-dimensional orthogonal group is defined by $\mathrm{O}(n) = \mathrm{Stiefel}(n, n)$. Applications of the orthogonal group include the Procrustes problem, which describes the optimal rotation and reflection transformations that best align one cloud of particles toward another (Doucet et al., 2001). The *special* orthogonal group is a subgroup of $\mathrm{O}(n)$ that satisfies the additional property that they have unit determinant. Formally, $\mathrm{SO}(n) \stackrel{\mathrm{def.}}{=} \mathrm{O}(n) \cap \{ \mathbf{M} \in \mathbb{R}^{n \times n} : \det(\mathbf{M}) = 1 \}$.

## 5 Theory

### 5.1 Change-of-Variables for Embedded Manifolds

For our theoretical development, the most important tool is the change-of-variables formula for embedded manifolds; see *inter alia* Rezende et al. (2020).

**Theorem 1** (change-of-variables formula for embedded manifolds)**.** Let $\mathcal{Y}$ and $\mathcal{Z}$ be smooth manifolds embedded in $\mathbb{R}^n$ and $\mathbb{R}^p$, respectively. Let $G : \mathbb{R}^m \to \mathcal{Y} \times \mathcal{Z}$ be a smooth, invertible transformation. Let $\pi_{\mathbb{R}^m}$ be a density on $\mathbb{R}^m$. Under the change-of-variables $G$, the corresponding density on $\mathcal{Y} \times \mathcal{Z}$ is given by,

$$\pi_{\mathcal{Y} \times \mathcal{Z}}(y, z) = \frac{\pi_{\mathbb{R}^m}(x)}{\sqrt{\det(\nabla G(x)^\top \nabla G(x))}} \tag{6}$$

where $x = G^{-1}(y, z)$.

It follows as a consequence that because $G$ is a diffeomorphism of non-empty manifolds that $\dim(\mathcal{Y} \times \mathcal{Z}) = \dim(\mathbb{R}^m) = m$. Even when $G$ is not an invertible mapping, it may be possible to compute the change-of-variables when $G$ is invertible on partitions of $\mathbb{R}^m$.

**Corollary 1.** Let $\mathcal{O}_1, \ldots, \mathcal{O}_l$ be a partition of $\mathbb{R}^m$. Let $G : \mathbb{R}^m \to \mathcal{Y} \times \mathcal{Z}$ be a function and suppose that there exist smooth and invertible functions $G_i : \mathcal{O}_i \to \mathcal{Y} \times \mathcal{Z}$ such that $G_i = G|\mathcal{O}_i$ for $i = 1, \ldots, l$. Then, if $x \sim \pi_{\mathbb{R}^m}$, the density of $(y, z) = G(x)$ is given by $\pi_{\mathcal{Y} \times \mathcal{Z}}(y, z) = \sum_{i=1}^{l} \frac{\pi_{\mathbb{R}^m}(x_i)}{\sqrt{\det(\nabla G_i(x_i)^\top \nabla G_i(x_i))}}$ where $x_i = G_i^{-1}(y, z)$.

How does theorem 1 relate to the dequantization of smooth manifolds? Manifolds (such the sphere, the torus, or the orthogonal group) can be introduced as elements of a new coordinate system for an ambient Euclidean space. By marginalizing out the other dimensions of the new coordinate system, we obtain the distribution on the manifold of interest. We have already seen an example of this in section 2 where $\mathbb{R}^3$ was transformed into a spherical coordinate system. Alternatively, given a manifold $\mathcal{Y}$, the auxiliary manifold $\mathcal{Z}$ is an additional structure such that we can find a diffeomorphism $G$ from $\mathbb{R}^m$ to $\mathcal{Y} \times \mathcal{Z}$ for some $m \in \mathbb{N}$. The auxiliary manifold can be thought of as facilitating the dequantization of $\mathcal{Y}$ into an ambient Euclidean space. We generalize this as follows:

#### 5.1.1 Example: Theorem 1 Applied to the Hypersphere

The hyperspherical coordinate transformation giving the isomorphism of $\mathbb{R}^m \setminus \{0\}$ and $\mathbb{S}^{m-1} \times \mathbb{R}_+$ is defined by, $x \mapsto (x/r, r)$ where $r = \|x\|_2$. The inverse transformation is $(s, r) \mapsto rs$. The Jacobian determinant of the hyperspherical coordinate transformation is $1/r^{m-1}$. Hence, given a density on $\mathbb{R}^m$ (equivalent to $\mathbb{R}^m \setminus \{0\}$ up to a set of Lebesgue measure zero) we may compute the change-of-variables so as to obtain a density on $\mathbb{S}^{m-1} \times \mathbb{R}_+$ by applying theorem 1 and the fact that the Jacobian determinant of the transformation is $1/r^{m-1}$. In other words, given a density $\pi_{\mathbb{R}^m}$ on $\mathbb{R}^m$, the manifold change-of-variables formula says that the corresponding density on $\mathrm{S}^{m-1} \times \mathbb{R}_+$ is

$$\pi_{\mathrm{S}^{m-1} \times \mathbb{R}_+}(s, r) = r^{m-1} \cdot \pi_{\mathbb{R}^m}(rs). \tag{7}$$

#### 5.1.2 Example: Theorem 1 Applied to the Torus

From the fact that $\mathbb{R}^2 \cong \mathbb{S}^1 \times \mathbb{R}_+$ (a consequence of the polar coordinate transformation), we may similarly develop a coordinate system of even-dimensional Euclidean space in which the torus appears. Recall that $\mathbb{T}^m \cong \mathbb{S}_1 \times \cdots \times \mathbb{S}^1$ (there are $m$ terms in the product). Therefore, we have established that $\mathbb{R}^{2m} \setminus \{0\} \cong \mathbb{T}^m \times \underbrace{\mathbb{R}_+ \times \cdots \times \mathbb{R}_+}_{m \text{ times}}$. The isomorphism can be explicitly constructed by writing $x = (x_{1,1}, x_{1,2}, \ldots, x_{m,1}, x_{m,2})$ and defining the map $G : \mathbb{R}^{2m} \setminus \{0\} \to \mathbb{T}^m \times \mathbb{R}_+ \times \ldots \times \mathbb{R}_+$ by $G(x) \overset{\text{def.}}{=} (x_1/r_1, \ldots, x_m/r_m, r_1, \ldots, r_m)$ where

$r_i = \sqrt{x_{i,1}^2 + x_{i,2}^2}$. As the concatenation of $m$ polar coordinate transformations, the Jacobian determinant of this transformation is $\prod_{i=1}^m r_i^{-1}$. The inverse transformation from $\mathbb{T}^m \times \mathbb{R}_+ \times \ldots \times \mathbb{R}_+$ back to $\mathbb{R}^{2m} \setminus \{0\}$ is given by $G^{-1}(s_1, \ldots, s_m, r_1, \ldots, r_m) = (r_1 s_1, \ldots, r_m s_m)$ where $(s_1, \ldots, s_m) \in \mathbb{S}^1 \times \ldots \times \mathbb{S}^1 \cong \mathbb{T}^m$. In the case of the torus, the product $\mathbb{R}_+ \times \ldots \times \mathbb{R}_+$ is the dequantization dimension of $\mathbb{T}^m$ into $\mathbb{R}^{2m} \setminus \{0\}$. In other words, in the case of $\mathbb{T}^2$, if one has a density $\pi_{\mathbb{R}^4} : \mathbb{R}^4 \to \mathbb{R}_+$ then the associated density under the change-of-variables $G$ is

$$\pi_{\mathbb{T}^2 \times \mathbb{R}_+ \times \mathbb{R}_+}(s_1, s_2, r_1, r_2) = r_1 r_2 \cdot \pi_{\mathbb{R}^4}((r_1 s_1, r_2 s_2)). \tag{8}$$

We describe a different means of dequantization, called *modulus dequantization*, in appendix C.

### 5.1.3  Example: Theorem 1 Applied to the Stiefel Manifold

The coordinate transformation in which the Stiefel manifold appears requires a more involved construction than was the case for the sphere or torus. We first recall the relationship between the Stiefel manifold, full-rank matrices (FR), and positive definite matrices (PD): $\mathrm{FR}(n,p) \cong \mathrm{Stiefel}(n,p) \times \mathrm{PD}(p)$. The isomorphism of these spaces is constructed by the *polar decomposition*. Given $\mathbf{M} \in \mathrm{FR}(n,p)$, define, $\mathbf{P} \overset{\text{def.}}{=} \sqrt{\mathbf{M}^\top \mathbf{M}} \in \mathrm{PD}(p)$ and $\mathbf{O} \overset{\text{def.}}{=} \mathbf{M}\mathbf{P}^{-1} \in \mathrm{Stiefel}(n,p)$. Thus, the isomorphism is given by $\mathbf{M} \mapsto (\mathbf{O}, \mathbf{P})$. This isomorphism is rigorously established in appendix D.

We further transform $\mathrm{PD}(n)$ in order to integrate over the manifold of positive-definite matrices. To do this we use the isomorphism of $\mathrm{PD}(n)$ and $\mathrm{Tri}_+(n)$, the set of $n \times n$ lower-triangular matrices with strictly positive entries on the diagonal. This isomorphism is known as the Cholesky decomposition: If $\mathbf{P} \in \mathrm{PD}(n)$ then there is a unique matrix $\mathbf{L} \in \mathrm{Tri}_+(n)$ such that $\mathbf{P} = \mathbf{L}\mathbf{L}^\top$. We have therefore proved $\mathrm{FR}(n,p) \cong \mathrm{Stiefel}(n,p) \times \mathrm{Tri}_+(p)$. One can use automatic differentiation in order to compute the Jacobian determinant of the transformation defined by $\mathbf{M} \mapsto (\mathbf{O}, \mathbf{L})$. We call the transformation $\mathbf{M} \mapsto (\mathbf{O}, \mathbf{L})$ the *Cholesky polar decomposition*. The inverse transformation is $(\mathbf{O}, \mathbf{L}) \mapsto \mathbf{O}\mathbf{L}\mathbf{L}^\top$

Finally we observe that almost all $n \times p$ matrices are full-rank. Therefore, if one has a density in $\mathbb{R}^{n \times p}$ then we may apply the polar decomposition coordinate transformation in order to obtain a density on $\mathrm{Stiefel}(n,p) \times \mathrm{PD}(p)$. Here $\mathrm{Tri}_+(p)$ plays the role of the dequantization dimension of $\mathrm{Stiefel}(n,p)$ into $\mathrm{FR}(n,p)$. In summary, given a density in $\pi_{\mathbb{R}^{n \times p}} : \mathbb{R}^{n \times p} \to \mathbb{R}_+$, we can construct the corresponding density on $\mathrm{Stiefel}(n,p) \times \mathrm{Tri}_+(p)$ by applying theorem 1:

$$\pi_{\mathrm{Stiefel}(n,p) \times \mathrm{Tri}_+(p)}(\mathbf{O}, \mathbf{L}) = \frac{\pi_{\mathbb{R}^{n \times p}}(\mathbf{OP})}{\mathcal{J}(\mathbf{OP})}, \tag{9}$$

where $\mathbf{P} = \mathbf{L}\mathbf{L}^\top$ and $\mathcal{J}(\mathbf{OP}) \overset{\text{def.}}{=} \sqrt{\det((\nabla G(\mathbf{OP}))^\top (\nabla G(\mathbf{OP})))}$.

An alternative coordinate transformation based on the QR decomposition instead of the polar decomposition is given in appendix J.

**Remark 1.** The Stiefel manifold $\mathrm{Stiefel}(n,p)$ is a generalization of the orthogonal group $\mathrm{O}(n)$ and we recover the latter exactly when $n = p$. In this case, we identify $\mathrm{FR}(n,n)$ as $\mathrm{GL}(n)$ so that we obtain $\mathrm{GL}(n) \cong \mathrm{O}(n) \times \mathrm{PD}(n)$.

## 5.2  Dequantization

We have now seen how several manifolds appear in coordinate systems. In each case, the manifold appears with an auxiliary manifold which may not be of immediate interest. Namely, (i) The sphere appears with set of positive real numbers when defining a coordinate system for $\mathbb{R}^m \setminus \{0\} \cong \mathbb{S}^{m-1} \times \mathbb{R}_+$; (ii) The torus appears the product manifold of $m$ copies of the positive real numbers when defining a coordinate system for $\mathbb{R}^{2m} \setminus \{0\} \cong \mathbb{T}^m \times \mathbb{R}_+ \times \ldots \times \mathbb{R}_+$; (iii) the Stiefel manifold appears with the set of lower-triangular matrices with positive diagonal entries when defining a coordinate system of $\mathrm{FR}(n,p) \cong \mathrm{Stiefel}(n,p) \times \mathrm{Tri}_+(p)$. We would like to marginalize out these "nuisance manifolds" so as to obtain distributions on the manifold of primary interest. A convenient means to achieve this is to introduce an importance sampling distribution

over the nuisance manifold. Formally, we have the following result, which is an immediate consequence of theorem 1.

**Corollary 2.** Let $\mathcal{Y}$, $\mathcal{Z}$, $G$, and $\pi_{\mathcal{Y} \times \mathcal{Z}}$ be as defined in theorem 1. Let $\tilde{\pi}_{\mathcal{Z}}$ be a non-vanishing density on $\mathcal{Z}$. To obtain the marginal density on $\mathcal{Y}$, let $x = G^{-1}(y, z)$ and it suffices to compute,

$$\pi_{\mathcal{Y}}(y) = \mathop{\mathbb{E}}_{z \sim \tilde{\pi}_{\mathcal{Z}}} \frac{\pi_{\mathcal{X}}(x)}{\tilde{\pi}_{\mathcal{Z}}(z) \cdot \sqrt{\det(\nabla G(x)^{\top} \nabla G(x))}}. \tag{10}$$

We refer to the auxiliary distribution $\tilde{\pi}_{\mathcal{Z}}$ as the *importance sampling distribution*. We consider some examples of marginalizing out the nuisance manifolds in some cases of interest.

### 5.2.1 Dequantization of the Hypersphere

Let $G : \mathbb{R}^m \to \mathbb{S}^{m-1} \times \mathbb{R}_+$ be the hyperspherical coordinate transformation described in section 5.1.1. To obtain the marginal distribution $\pi_{\mathbb{S}^{m-1}}$ we compute, $\pi_{\mathbb{S}^{m-1}}(s) = \int_{\mathbb{R}_+} r^{m-1} \cdot \pi_{\mathbb{R}^m}(rs) \, dr$. Or, using an importance sampling distribution $\tilde{\pi}_{\mathbb{R}_+}$ whose support is $\mathbb{R}_+$, substituting (7) into (10) we obtain, $\pi_{\mathbb{S}^{m-1}}(s) = \mathop{\mathbb{E}}_{r \sim \tilde{\pi}_{\mathbb{R}_+}} \frac{r^{m-1} \cdot \pi_{\mathbb{R}^m}(rs)}{\tilde{\pi}_{\mathbb{R}_+}(r)}$.

### 5.2.2 Dequantization of the Torus

Let $G$ be the toroidal coordinate transformation described in section 5.1.2. Similar to section 5.2.1, one can introduce an importance sampling distribution on $\mathbb{R}_+ \times \mathbb{R}_+$ so as to obtain the marginal distribution on the torus as $\pi_{\mathbb{T}^2}(s_1, s_2) = \mathop{\mathbb{E}}_{r_1, r_2 \sim \tilde{\pi}_{\mathbb{R}_+ \times \mathbb{R}_+}} \frac{\pi_{\mathbb{R}^4}((r_1 s_1, r_2 s_2))}{\tilde{\pi}_{\mathbb{R}_+ \times \mathbb{R}_+}(r_1, r_2)/r_1 r_2}$.

### 5.2.3 Dequantization of the Stiefel Manifold

Let $G$ be the Cholesky polar decomposition coordinate transformation described in section 5.1.3. To construct an importance sampling distribution over $\mathrm{Tri}_+(p)$, one could generate the diagonal entries of $\mathbf{L}_{ii} \sim \mathrm{LogNormal}(\mu_i, \sigma_i^2)$ and the remaining entries in the lower triangle according to $\mathbf{L}_{ij} \sim \mathrm{Normal}(\mu_{ij}, \sigma_{ij}^2)$. Applying corollary 2 gives the importance sampling formula for the marginal density on $\mathrm{Stiefel}(n, p)$:

$$\pi_{\mathrm{Stiefel}(n,p)}(\mathbf{O}) = \mathop{\mathbb{E}}_{\mathbf{L} \sim \tilde{\pi}_{\mathrm{Tri}_+}} \frac{\pi_{\mathbb{R}^{n \times p}}(\mathbf{OP})}{\mathcal{J}(\mathbf{OP}) \cdot \tilde{\pi}_{\mathrm{Tri}_+}(\mathbf{L})} \tag{11}$$

where $\mathbf{P} = \mathbf{LL}^T$.

## 6 Proposed Method

We investigate the problem of density estimation given observations on a manifold using the dequantization procedure described in section 5. Let $\mathcal{Y}$ be a manifold embedded in $\mathbb{R}^n$ and let $\pi_{\mathcal{Y}}$ be a density on $\mathcal{Y}$. Given observations of $\pi_{\mathcal{Y}}$, we wish to construct an estimate $\hat{\pi}_{\mathcal{Y}}$ of the density $\pi_{\mathcal{Y}}$. Algorithm 1 shows how we may apply dequantization for the purposes of density estimation provided that we have samples from the target density. We apply eq. (10) in order to obtain the density estimate on $\mathcal{Y}$. Generating samples from $\pi_{\mathcal{Y}}$ may be achieved by first sampling $x \sim \pi_{\mathcal{X}}$, applying the transformation $G(x) = (y, z)$, and taking $y$ as a sample from the approximated distribution $\hat{\pi}_{\mathcal{Y}}$.

**Densities on $\mathbb{R}^m$.** As $\mathbb{R}^m$ is a Euclidean space, we have available a wealth of possible mechanisms to produce flexible densities in the ambient space. One popular choice is RealNVP (Dinh et al., 2017). In this case $\theta$ is the parameters of the underlying RealNVP network. An alternative is neural ODEs wherein $\theta$ parameterizes a vector field in the Euclidean space; the change in probability density under the vector field flow is obtained by integrating the instantaneous change-of-variables formula (Chen et al., 2018; Grathwohl et al., 2018).

---

**Algorithm 1** Training loop for dequantization inference. The target density $\pi_\mathcal{Y}$ is only relevant insofar as we must have samples available from it. The algorithm produces $\theta$ and $\phi$ that parameterize a distribution on $\mathbb{R}^m$ and a dequantization density on $\mathcal{Z}$. Together, these two densities can be combined to compute an estimate of the density using the right-hand side of eq. (10).

---

1: **Input**: Samples from target density $\mathcal{D} \stackrel{\text{def.}}{=} (y_1, \ldots, y_B)$ on an embedded manifold $\mathcal{Y} \subset \mathbb{R}^m$, step-size $\epsilon \in \mathbb{R}_+$.
2: Identify a smooth change-of-variables $G : \mathbb{R}^m \to \mathcal{Y} \times \mathcal{Z}$ where $\mathcal{Z} \subset \mathbb{R}^p$ is an auxiliary structure.
3: Let $\pi_{\mathbb{R}^m}$ be a density on $\mathbb{R}^m$ that is smoothly parameterized by $\theta \in \mathbb{R}^{n_{\text{amb}}}$.
4: Let $\tilde{\pi}_\mathcal{Z}$ be a density on $\mathcal{Z}$ that is continuously parameterized by $y \in \mathcal{Y}$ and smoothly parameterized by $\phi \in \mathbb{R}^{n_{\text{deq}}}$.
5: **while** Not Done **do**
6:     Use eq. (13) or eq. (14) to compute an average loss over $\mathcal{D}$:

$$\mathcal{L}(\theta, \phi | \mathcal{D}) = \mathop{\mathbb{E}}_{y \sim \text{Unif}(\mathcal{D})} \mathcal{F}(y | \theta, \phi) \tag{12}$$

   where $\mathcal{F}$ is the right-hand side of eq. (13) or eq. (14), respectively.
7:     Update $\theta = \theta + \epsilon \nabla_\theta \mathcal{L}(\theta, \phi | \mathcal{D})$ and $\phi = \phi + \epsilon \nabla_\phi \mathcal{L}(\theta, \phi | \mathcal{D})$.
8: **end while**
9: **Output**: Parameterized densities $\pi_{\mathbb{R}^m}(\cdot | \theta)$ and $\tilde{\pi}_\mathcal{Z}(\cdot | y, \phi)$ that can be used to perform density estimation on $\mathcal{Y}$ using eq. (10).

---

**Objective Functions.** We consider two possible objective functions for density estimation. The first is the evidence lower bound of the observations $\{y_1, \ldots, y_{n_{\text{obs}}}\}$:

$$\log \hat{\pi}_\mathcal{Y}(y_i) \geq \mathop{\mathbb{E}}_{z \sim \tilde{\pi}_\mathcal{Z}} \log \frac{\pi_{\mathbb{R}^m}(G^{-1}(y_i, z))}{\tilde{\pi}_\mathcal{Z}(z) \cdot \sqrt{\det(\nabla G(x)^\top \nabla G(x))}}. \tag{13}$$

This follows as a consequence of Jensen's inequality applied to eq. (10). Experimental results using this objective function are denoted with the suffix (ELBO). The second is the log-likelihood computed via importance sampling:

$$\log \hat{\pi}_\mathcal{Y}(y_i) = \log \mathop{\mathbb{E}}_{z \sim \tilde{\pi}_\mathcal{Z}} \frac{\pi_{\mathbb{R}^m}(G^{-1}(y_i, z))}{\tilde{\pi}_\mathcal{Z}(z) \cdot \sqrt{\det(\nabla G(x)^\top \nabla G(x))}}. \tag{14}$$

Because the calculation of eq. (14) requires an importance sampling estimate, experimental results using this objective function are denoted with the suffix (I.S.).

## 7 Experimental Results

To demonstrate the effectiveness of the approach, we now show experimental results for density estimation on three different manifolds: the sphere, the torus and the orthogonal group. In our comparison against competing algorithms, we ensure that each method has a comparable number of learnable parameters. Our evaluation metrics are designed to test the fidelity of the density estimate to the target distribution; details on evaluation metrics are given in appendix G. We note that lower is better for all evaluation metrics except for the relative effective sample size (ESS) for which larger values (closer to 100) are better. In all of our examples we use rejection sampling in order to draw samples from the target distribution.

**Sphere and Hypersphere.** Our first experimental results concern the sphere $\mathbb{S}^2$ where we consider a multimodal distribution with four modes. This density on $\mathbb{S}^2$ which is visualized in fig. 3. We consider performing density estimation using the ELBO (eq. (13)) and log-likelihood objective functions (eq. (14)); we construct densities in the ambient space using RealNVP and neural ODEs. We use algorithm 1 to learn the parameters of the ambient and dequantization distributions. As baselines we consider the Möbius transform approach described in Rezende et al. (2020), which is a specialized normalizing flow method for

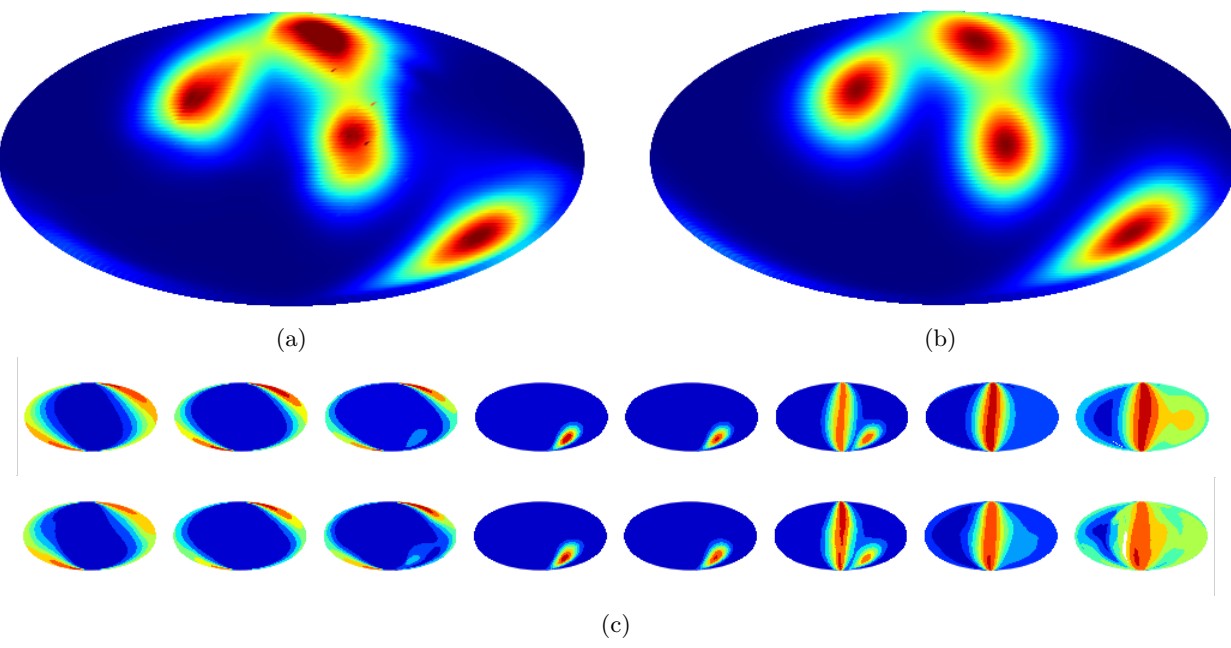

(a)                                                                    (b)

(c)

Figure 3: Figures (a) and (b): Comparison of the density on $\mathbb{S}^2$ learned via RealNVP dequantization with a KL-divergence loss estimated via importance sampling. The dequantization procedure (a) is able to identify all modes of the true density (b) correctly. Figure (c): Comparison of the density on $\mathbb{S}^3$ learned via RealNVP dequantization with a KL-divergence loss. We visualize $\mathbb{S}^2$-slices of $\mathbb{S}^3$ by examining the density when one of the hyperspherical coordinates is held fixed at different values. The top set of slices shows the approximate density obtained via dequantization and the target density and the bottom slices show the true density. The KL divergence is 0.01269 and the relative effective sample size is 97.70%.

Table 1: Comparison of dequantization to normalizing flows on the multimodal density on $\mathbb{S}^2$. Averages were computed using ten random trials for the dequantization procedures and eight random trials for the normalizing flow (because two random trials exhibited divergent behavior and were excluded). The dequantization procedure is illustrated for both the ELBO loss and the KL divergence loss.

| Method | Mean MSE | Covariance MSE | KL($q\|p$) | KL($p\|q$) | Relative ESS |
|---|---|---|---|---|---|
| Deq. ODE (ELBO) | 0.0012 ± 0.0002 | 0.0006 ± 0.0001 | 0.0046 ± 0.0002 | 0.0046 ± 0.0002 | 99.0990 ± 0.0401 |
| Deq. ODE (I.S.) | 0.0014 ± 0.0002 | 0.0010 ± 0.0001 | 0.0029 ± 0.0001 | 0.0029 ± 0.0001 | 99.4170 ± 0.0225 |
| Deq RealNVP (ELBO) | 0.0004 ± 0.0001 | 0.0003 ± 0.0001 | 0.0231 ± 0.0010 | 0.0212 ± 0.0009 | 95.9540 ± 0.1688 |
| Deq. RealNVP (I.S.) | 0.0005 ± 0.0002 | 0.0002 ± 0.0000 | 0.0124 ± 0.0006 | 0.0115 ± 0.0006 | 97.8240 ± 0.1183 |
| Man. ODE | 0.0010 ± 0.0004 | 0.0009 ± 0.0002 | 0.0085 ± 0.0007 | 0.0083 ± 0.0007 | 98.3860 ± 0.1328 |
| Möbius | 0.0021 ± 0.0005 | 0.0019 ± 0.0005 | 0.0595 ± 0.0025 | — | 89.2575 ± 0.4888 |

tori and spheres, and the neural manifold ODE applied to the sphere as described in (Lou et al., 2020). We give a comparison of performance metrics between these methods in table 1. In these experiments, we find that parameterizing a neural ODE model in the ambient space gave the better KL-divergence and effective sample size (ESS) metrics than RealNVP when our dequantization approach is used. We find that our dequantization algorithm minimizing either eq. (13) or eq. (14) achieves similar performance in the first and second moment metrics. However, when using eq. (14), slightly lower KL-divergence metrics are achievable as well as slightly larger effective sample sizes. In either case, dequantization outperforms the Möbius transform on this multimodal density on $\mathbb{S}^2$. The manifold ODE method is outperformed by the ODE dequantization algorithms with both eq. (13) and eq. (14).

We next consider a multimodal density $\mathbb{S}^3 \cong SU(3)$ (the special unitary group). As before, we compare dequantization to Möbius flow transformations and manifold neural ODEs and present results in table 2.

Table 2: Comparison of dequantization to normalizing flows on the multimodal density on $\mathbb{S}^3$. Averages were computed using ten random trials for the dequantization procedures and nine random trials for the normalizing flow (one random trial exhibited divergent behavior and was excluded).

| Method | Mean MSE | Covariance MSE | KL$(q\|p)$ | KL$(p\|q)$ | Relative ESS |
|---|---|---|---|---|---|
| Deq. ODE (ELBO) | $0.0009 \pm 0.0001$ | $0.0007 \pm 0.0001$ | $0.0072 \pm 0.0002$ | $0.0070 \pm 0.0002$ | $98.6490 \pm 0.0388$ |
| Deq. ODE (I.S.) | $0.0017 \pm 0.0001$ | $0.0022 \pm 0.0002$ | $0.0189 \pm 0.0004$ | $0.0180 \pm 0.0004$ | $96.6150 \pm 0.0648$ |
| Deq. RealNVP (ELBO) | $0.0003 \pm 0.0001$ | $0.0004 \pm 0.0001$ | $0.0384 \pm 0.0010$ | $0.0283 \pm 0.0005$ | $95.1880 \pm 0.0771$ |
| Deq. RealNVP (I.S.) | $0.0003 \pm 0.0001$ | $0.0003 \pm 0.0000$ | $0.0208 \pm 0.0004$ | $0.0180 \pm 0.0004$ | $96.6340 \pm 0.0920$ |
| Man. ODE | $0.0012 \pm 0.0003$ | $0.0008 \pm 0.0002$ | $0.0098 \pm 0.0009$ | $0.0094 \pm 0.0007$ | $98.1780 \pm 0.1302$ |
| Möbius | $0.0027 \pm 0.0004$ | $0.0014 \pm 0.0003$ | $0.0542 \pm 0.0047$ | — | $88.7290 \pm 0.9332$ |

Table 3: Comparison of Deq. RealNVP to Möbius Flow flows on the multimodal density on $\mathbb{T}^2$. Averages were computed using ten random trials for the Deq. RealNVP procedures, direct, and Möbius Flow flow procedures.

| Density | Method | Mean MSE | Covariance MSE | KL$(q\|p)$ | KL$(p\|q)$ | Relative ESS |
|---|---|---|---|---|---|---|
| Correlated | Deq. RealNVP (ELBO) | $0.0019 \pm 0.0007$ | $0.0098 \pm 0.0019$ | $0.0072 \pm 0.0005$ | $0.0084 \pm 0.0006$ | $98.6510 \pm 0.0958$ |
| | Deq. RealNVP (I.S.) | $0.0024 \pm 0.0007$ | $0.0121 \pm 0.0041$ | $0.0038 \pm 0.0004$ | $0.0040 \pm 0.0004$ | $99.2570 \pm 0.0796$ |
| | Modulus | $0.0017 \pm 0.0005$ | $0.0072 \pm 0.0021$ | $0.0025 \pm 0.0006$ | $0.0025 \pm 0.0006$ | $99.5000 \pm 0.1153$ |
| | Möbius | $0.0004 \pm 0.0002$ | $0.0083 \pm 0.0029$ | $0.0021 \pm 0.0002$ | — | $99.5850 \pm 0.0332$ |
| Unimodal | Deq. RealNVP (ELBO) | $0.0010 \pm 0.0003$ | $0.0133 \pm 0.0040$ | $0.0066 \pm 0.0004$ | $0.0079 \pm 0.0004$ | $98.7240 \pm 0.0761$ |
| | Deq. RealNVP (I.S.) | $0.0011 \pm 0.0003$ | $0.0081 \pm 0.0020$ | $0.0021 \pm 0.0002$ | $0.0023 \pm 0.0002$ | $99.5980 \pm 0.0317$ |
| | Modulus | $0.0014 \pm 0.0003$ | $0.0164 \pm 0.0028$ | $0.0028 \pm 0.0003$ | $0.0028 \pm 0.0003$ | $99.4340 \pm 0.0531$ |
| | Möbius | $0.0006 \pm 0.0002$ | $0.0055 \pm 0.0037$ | $0.0008 \pm 0.0001$ | — | $99.8490 \pm 0.0293$ |
| Multimodal | Deq. RealNVP (ELBO) | $0.0024 \pm 0.0006$ | $0.0158 \pm 0.0026$ | $0.0065 \pm 0.0002$ | $0.0075 \pm 0.0002$ | $98.7800 \pm 0.0379$ |
| | Deq. RealNVP (I.S.) | $0.0007 \pm 0.0002$ | $0.0061 \pm 0.0014$ | $0.0020 \pm 0.0001$ | $0.0022 \pm 0.0002$ | $99.6160 \pm 0.0288$ |
| | Modulus | $0.0016 \pm 0.0007$ | $0.0063 \pm 0.0024$ | $0.0035 \pm 0.0004$ | $0.0035 \pm 0.0004$ | $99.3030 \pm 0.0725$ |
| | Möbius | $0.0006 \pm 0.0001$ | $0.0070 \pm 0.0039$ | $0.0012 \pm 0.0002$ | — | $99.7600 \pm 0.0358$ |

Similar to the case of the multimodal density on $\mathbb{S}^2$, we find that dequantization with an ambient neural ODE model is most effective, with ELBO maximization giving the smallest KL-divergence metrics. All dequantization algorithms out-performed the Möbius transformation on the sphere but only dequantization with an ambient ODE and ELBO minimization outperformed the manifold neural ODE method.

**Torus.** We next consider three densities from Rezende et al. (2020) on the torus $\mathbb{T}^2$. These densities are, respectively, unimodal, multimodal, or exhibit strongly correlated dimensions. As in the case of the sphere, we evaluate dequantization of the torus against the Möbius transform method. As a further point of comparison, we also consider learning a normalizing flow density on $\mathbb{R}^2$ and simply identifying every $2\pi$-periodic point so as to induce distribution on $\mathbb{T}^2$; we call this the modulus dequantization method. Results are reported in table 3. We find that the Möbius transformation performs strongest in this comparison. The modulus method also performs well, particularly on the correlated toroidal density function. Of the dequantization-based approaches, minimizing the negative log-likelihood produces the best performance, which outperforms the modulus method in terms of KL-divergence and first- and second-moment metrics (excepting the correlated density). We note, however, that all of these methods estimated effective sample sizes at nearly 100%, indicating that the differences between each approach, while statistically significant, are practically marginal.

**Orthogonal Group.** The previous two examples focused on manifolds composed of spheres and circles. We now examine density estimation on the orthogonal group, where we consider inference in a probabilistic variant of the orthogonal Procrustes problem; we seek to sample orthogonal transformations that transport one point cloud towards another in terms of squared distance. We consider parameterizing a distribution in the ambient Euclidean space using RealNVP in these experiments. Results are presented in table 4. We observe that optimizing the ELBO objective function (eq. (13)) tended to produce better density estimates than the log-likelihood (eq. (14)). Nevertheless, we find that either dequantization algorithm is effective at matching the target density.

Table 4: Metrics of the dequantization algorithm in application to the orthogonal Procrustes problem and dequantization of a multimodal density on SO(3). When using the polar decomposition, results are averaged over ten independent trials for the multimodal distribution on SO(3) and nine independent trials for the orthogonal Procrustes problem.

| Experiment | Mean MSE | Covariance MSE | $\text{KL}(q\|p)$ | $\text{KL}(p\|q)$ | Relative ESS |
|---|---|---|---|---|---|
| Procrustes (ELBO) | $0.0022 \pm 0.0013$ | $0.0014 \pm 0.0009$ | $0.0159 \pm 0.0043$ | $0.0122 \pm 0.0030$ | $97.1000 \pm 0.6099$ |
| Procrustes (I.S.) | $0.0032 \pm 0.0017$ | $0.0016 \pm 0.0010$ | $0.0220 \pm 0.0070$ | $0.0155 \pm 0.0045$ | $96.7322 \pm 0.7752$ |
| SO(3) (ELBO) | $0.0005 \pm 0.0001$ | $0.0024 \pm 0.0002$ | $0.0459 \pm 0.0024$ | $0.0357 \pm 0.0024$ | $96.3050 \pm 0.1004$ |
| SO(3) (I.S.) | $0.0007 \pm 0.0003$ | $0.0011 \pm 0.0003$ | $0.0198 \pm 0.0025$ | $0.0230 \pm 0.0022$ | $97.7550 \pm 0.1588$ |

We may also leverage corollary 1 so as to apply our method to the "dequantization" of SO($n$). As an example, we consider a multimodal density on SO(3). Results of applying our method to sampling from this distribution are also shown in table 4. In this example we find that minimizing the negative log-likelihood using importance sampling tended to produce the best approximation of the first- and second-moments of the distribution, in addition to smaller KL-divergence metrics.

## 8 Conclusion

This paper proposed a new method for density estimation on manifolds called manifold dequantization. The proposed approach allows us to make use of existing techniques for density estimation on Euclidean spaces while still providing efficient, exact sampling of the distribution on the manifold as well as approximate density calculation. We evaluated this method for densities on the sphere, the torus, and the orthogonal group. Our results show that manifold dequantization is competitive with, or exceeds the performance of, competing methods for density estimation on manifolds.

### Broader Impact Statement

In terms of societal impact, matrix manifolds appear in various scientific disciples such as computational biology, the earth sciences, and robotics. While our experimentation on dequantization are synthetic, we expect that the techniques proposed here can be adapted to research in these scientific disciplines.

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

# A   Evidence Lower Bound for Special Orthogonal Group

Let $\mathbf{R}$ be a reflection matrix. Then using the fact that $\mathrm{O}(n) = \mathrm{SO}(n) \times \{\mathrm{Id}_n, \mathbf{R}\}$ we have,

$$\log \pi_{\mathrm{SO}(n)}(\mathbf{O}) = \log \underset{\mathbf{S} \sim \mathrm{Unif}(\mathrm{Id}_n, \mathbf{R})}{\mathbb{E}} \pi_{\mathrm{O}(n)}(\mathbf{SO}) + \log 2 \tag{15}$$

$$\geq \underset{\mathbf{S} \sim \mathrm{Unif}(\mathrm{Id}_n, \mathbf{R})}{\mathbb{E}} \log \pi_{\mathrm{O}(n)}(\mathbf{SO}) + \log 2 \tag{16}$$

$$\geq \underset{\mathbf{S} \sim \mathrm{Unif}(\mathrm{Id}_n, \mathbf{R})}{\mathbb{E}} \underset{\mathbf{L} \sim \tilde{\pi}_{\mathrm{Tri}_+}}{\mathbb{E}} \log \frac{\pi_{\mathbb{R}^{n \times n}}(\mathbf{SOP})}{\mathcal{J}(\mathbf{OP}) \cdot \tilde{\pi}_{\mathrm{Tri}_+}(\mathbf{L})} + \log 2 \tag{17}$$

$$\geq \underset{\mathbf{S} \sim \mathrm{Unif}(\mathrm{Id}_n, \mathbf{R})}{\mathbb{E}} \underset{\mathbf{L} \sim \tilde{\pi}_{\mathrm{Tri}_+}}{\mathbb{E}} \log \frac{\pi_{\mathbb{R}^{n \times n}}(\mathbf{SOP})}{\mathcal{J}(\mathbf{OP}) \cdot \tilde{\pi}_{\mathrm{Tri}_+}(\mathbf{L})} \tag{18}$$

where $\mathcal{J}(\mathbf{OP}) \overset{\mathrm{def.}}{=} \sqrt{\det((\nabla G(\mathbf{OP}))^\top (\nabla G(\mathbf{OP})))}$.

## B  Dequantizing the Integers

We now consider how to apply corollary 1 and theorem 1 to the dequantization of integers and draw a connection to dequantization in normalizing flows (Hoogeboom et al., 2020). Let $\pi_{\mathbb{R}}$ be a probability density on $\mathbb{R}$. Consider the function $T : \mathbb{R} \to \mathbb{Z} \times [0, 1)$ defined by $T(x) = (\lfloor x \rfloor, x - \lfloor x \rfloor)$. Consider the partition of $\mathbb{R}$ given by $\mathcal{O}_n = [n, n + 1)$; on each $\mathcal{O}_n$ we have that $T_n(x) \overset{\text{def.}}{=} (n, x - n)$ satisfies $T_n = T | \mathcal{O}_n$; moreover, on each $\mathcal{O}_n$, the transformation $T_n$ is invertible (the inverse map is $(n, r) \mapsto n + r$) and preserves volume, which implies a unit Jacobian determinant. Therefore, the associated density on $\mathbb{Z} \times [0, 1)$ is given by $\pi_{\mathbb{Z} \times [0,1)}(n, r) = \pi_{\mathbb{R}}(n + r)$. We may view $\mathbb{Z} \times [0, 1)$ as a coordinate system for $\mathbb{R}$. If we wish to integrate out the nuisance variables in the unit interval, we obtain the marginal density on $\mathbb{Z}$ as

$$\pi_{\mathbb{Z}}(n) = \int_0^1 \pi_{\mathbb{R}}(n + r) \, \mathrm{d}r. \tag{19}$$

By choosing $\tilde{\pi}_{[0,1)}$ as, for example, a Uniform distribution, one obtains the traditional uniform dequantization. Choosing a more complex distribution, for example, a Beta distribution (with parameters possibly depending on $n$) one obtains an importance sampling formula for the marginal density as $\pi_{\mathbb{Z}}(n) = \underset{r \sim \tilde{\pi}_{[0,1)}(r)}{\mathbb{E}} \frac{\pi_{\mathbb{R}}(n+r)}{\tilde{\pi}_{[0,1)}(r)}$.

This can be optimized through a variational bound, giving variational dequantization (Ho et al., 2019). Optimizing it directing gives importance weighted dequantization (Hoogeboom et al., 2020).

## C  Torus Modulus Dequantization

Taking inspiration from appendix B, we turn now to consider another dequantization of the torus. To begin, identify the circle $\mathbb{S}^1$ with the interval $[0, 2\pi)$. We can construct a map from $\mathbb{R}$ to $\mathbb{Z} \times [0, 2\pi)$ by identifying $2\pi$-periodic points. Define the map $\tilde{G}(x) = (\lfloor x \rfloor, x \pmod{2\pi})$. Let $\mathcal{O}_k = [2\pi k, 2\pi(k + 1))$ for $k \in \mathbb{Z}$; on this interval we may define $G_k : \mathcal{O}_k \to \mathbb{Z} \times [0, 2\pi)$ by $G_k(x) = (k, x - 2\pi k)$ which satisfies $G|\mathcal{O}_k = G_k$. Because this map is nothing but a shift by $2\pi k$, it is invertible and volume preserving. Moreover, we may view $\mathbb{Z} \times [0, 2\pi)$ as a coordinate system for $\mathbb{R}$. Let $\pi_{\mathbb{R}}$ be a density on $\mathbb{R}$ and let $x \sim \pi_{\mathbb{R}}$. The associated density on $\mathbb{Z} \times [0, 2\pi)$ is therefore,

$$\pi_{\mathbb{Z} \times [0,2\pi)}(k, y) = \pi_{\mathbb{R}}(y + 2\pi k). \tag{20}$$

We may marginalize out the integers to obtain the density on $[0, 2\pi)$ as

$$\pi_{[0,2\pi)}(y) = \sum_{k \in \mathbb{Z}} \pi_{\mathbb{R}}(y + 2\pi k). \tag{21}$$

This idea is readily extended to higher dimensions. Because the torus is nothing but the product manifold of two circles, we may identify the torus with $[0, 2\pi) \times [0, 2\pi)$. Let $\pi_{\mathbb{R}^2}$ be a density on $\mathbb{R}^2$ and let $G : \mathbb{R}^2 \to [\mathbb{Z} \times [0, 2\pi)]^2$ be defined by $G(x_1, x_2) = (\tilde{G}(x_1), \tilde{G}(x_2))$. Following the precise reasoning from the one-dimensional case, if $x \sim \pi_{\mathbb{R}^2}$ then the density of $(x_1 \pmod{2\pi}, x_2 \pmod{2\pi})$ is,

$$\pi_{[0,2\pi) \times [0,2\pi)}(y_1, y_2) = \sum_{k_1 \in \mathbb{Z}} \sum_{k_2 \in \mathbb{Z}} \pi_{\mathbb{R}^2}(y_1 + 2\pi k_1, y_2 + 2\pi k_2). \tag{22}$$

We call this approach *modulus dequantization*. This approach requires fewer dimensions than embedding the torus in $\mathbb{R}^{2m}$. In practice, the infinite sums over the integers may be approximated by truncating to a finite number of terms.

## D  Polar Decomposition Isomorphism

**Proposition 1.** Let $\mathbf{A}$ be a real, positive semi-definite matrix. Then there exists a unique real, positive semi-definite matrix $\mathbf{B}$ such that $\mathbf{A} = \mathbf{BB}$. We call $\mathbf{B}$ the principal square root of $\mathbf{A}$ and may write $\mathbf{B} = \sqrt{\mathbf{A}}$.

**Definition 1** (Stiefel Manifold)**.** The Stiefel$(m, n)$ manifold is the subset of $\mathbb{R}^{m \times n}$ of orthogonal matrices. That is,

$$\text{Stiefel}(m, n) = \left\{ \mathbf{X} \in \mathbb{R}^{m \times n} : \mathbf{X}^\top \mathbf{X} = \text{Id} \right\} \tag{23}$$

**Definition 2.** The set of $n \times n$ positive-definite matrices is denoted $\text{PD}(n)$.

**Definition 3** (Non-Square Polar Decomposition)**.** Let $\mathbf{A} \in \mathbb{R}^{m \times n}$ with $m \geq n$ have the (thin) singular value decomposition $\mathbf{A} = \mathbf{U\Sigma V}^\top$ where $\mathbf{U} \in \text{Stiefel}(m, n)$ and $\mathbf{V}^\top \in \text{O}(n)$. Then we define the non-square polar decomposition to be $\mathbf{A} = \mathbf{OP}$ where,

$$\mathbf{O} \stackrel{\text{def.}}{=} \mathbf{UV}^\top \tag{24}$$

$$\mathbf{P} \stackrel{\text{def.}}{=} \mathbf{V\Sigma V}^\top. \tag{25}$$

**Lemma 1.** The quantity $\mathbf{P}$ in eq. (25) is uniquely defined.

*Proof.* The strategy is to show that $\mathbf{P}$ is the unique principal square root of a positive semi-definite matrix. It is immediately clear from the definition that $\mathbf{P}$ is itself positive semi-definite. Consider the positive semi-definite matrix

$$\mathbf{A}^\top \mathbf{A} = (\mathbf{V\Sigma U}^\top)(\mathbf{U\Sigma V}^\top) \tag{26}$$

$$= \mathbf{V\Sigma\Sigma V}^\top \tag{27}$$

$$= (\mathbf{V\Sigma V}^\top)(\mathbf{V\Sigma V}^\top) \tag{28}$$

$$= \mathbf{PP}. \tag{29}$$

By identification $\mathbf{P}$ is the principal square root of $\mathbf{A}^\top \mathbf{A}$ so it is unique. $\square$

**Proposition 2.** Let $\Sigma = \text{diag}(\sigma_1, \ldots, \sigma_n)$ and suppose that $\sigma_1 \geq \sigma_2 \geq \ldots \geq \sigma_n > 0$. Then, the quantity $\mathbf{O}$ in eq. (24) is uniquely defined.

*Proof.* If $\sigma_n > 0$ then

$$\det(\mathbf{P}) = \det(\mathbf{V}) \cdot \det(\Sigma) \cdot \det(\mathbf{V}^\top) \tag{30}$$

$$= \prod_{i=1}^{n} \sigma_i \tag{31}$$

$$> 0. \tag{32}$$

Thus, $\mathbf{P}$ is invertible. Thus, $\mathbf{O} = \mathbf{AP}^{-1}$. $\square$

Note that the condition on the singular values is equivalent to the statement that $\mathbf{A}$ has full-rank.

**Lemma 2.** If $\mathbf{A}$ has full-rank then $\mathbf{P} \in \text{PD}(n)$.

*Proof.* If $\mathbf{A}$ has full-rank then all the singular values are strictly positive. Thus, $\mathbf{V\Sigma V}^\top$ is an eigen-decomposition of $\mathbf{P}$ whose eigenvalues are all positive. Since a matrix is positive-definite if and only if all of its eigenvalues are positive, we conclude that $\mathbf{P} \in \text{PD}(n)$. $\square$

**Lemma 3.** The quantity $\mathbf{O}$ in eq. (24) is an element of Stiefel$(m, n)$.

*Proof.*

$$\mathbf{O}^\top \mathbf{O} = \mathbf{V}\mathbf{U}^\top \mathbf{U}\mathbf{V}^\top \tag{33}$$

$$= \mathbf{V}\mathbf{V}^\top \tag{34}$$

$$= \mathrm{Id}. \tag{35}$$

$\square$

**Proposition 3.** Given $\mathbf{O} \in \mathrm{Stiefel}(m, n)$ and $\mathbf{P} \in \mathrm{PD}(n)$, we may always write $\mathbf{O}$ and $\mathbf{P}$ in the form of eqs. (24) and (25).

*Proof.* Since $\mathbf{P} \in \mathrm{PD}(n)$, by the spectral theorem there exists an orthonormal basis $\{v_1, \ldots, v_n\}$ of $\mathbb{R}^n$ and real, positive eigenvalues $\lambda_1, \ldots, \lambda_n$, such that $\mathbf{P} = \mathbf{V}\Sigma\mathbf{V}^{-1} = \mathbf{V}\Sigma\mathbf{V}^\top$ where $\mathbf{V} \in \mathbb{R}^{n \times n}$ is the collection of $\{v_1, \ldots, v_n\}$ as columns and $\Sigma = \mathrm{diag}(\lambda_1, \ldots, \lambda_n)$.

Using this $\mathbf{V}$ we may compute $\mathbf{U} = \mathbf{O}\mathbf{V}$, which is an orthogonal matrix:

$$\mathbf{U}^\top \mathbf{U} = \mathbf{V}^\top \mathbf{O}^\top \mathbf{O}\mathbf{V} \tag{36}$$

$$= \mathbf{V}^\top \mathbf{V} \tag{37}$$

$$= \mathrm{Id}. \tag{38}$$

Define the matrix $\mathbf{A} = \mathbf{O}\mathbf{P}$. By inspection, a (thin) singular value decomposition of $\mathbf{A}$ is

$$\mathbf{A} = \mathbf{U}\Sigma\mathbf{V}^\top \tag{39}$$

since

$$\mathbf{O}\mathbf{P} = \mathbf{U}\mathbf{V}^\top \mathbf{V}\Sigma\mathbf{V}^\top \tag{40}$$

$$= \mathbf{U}\Sigma\mathbf{V}^\top. \tag{41}$$

Finally, $\mathbf{U} \in \mathrm{Stiefel}(m, n)$, $\Sigma$ has only positive entries, and $\mathbf{V} \in \mathrm{O}(n)$, which are the conditions of a thin singular value decomposition. $\square$

# E Experimental Details

Here we include additional information about our experimental design.

## E.1 Sphere

We consider the following unnormalized density on $\mathbb{S}^2$ given by

$$\pi_{\mathbb{S}^2}(y) \propto \sum_{i=1}^{4} \exp(10 y^\top \mu_i) \tag{42}$$

where $\mu_1 = (0.763, 0.643, 0.071)$, $\mu_2 = (0.455, -0.708, 0.540)$, $\mu_3 = (0.396, 0.271, 0.878)$, and $\mu_4 = (-0.579, 0.488, -0.654)$.

When using RealNVP, the total number of learnable parameters in our dequantization model is $5,894$; dequantization with an ambient ODE has $5,705$ learnable parameters; in the case of the Möbius transform the total number of learnable parameters is $5,943$; for the manifold ODE implementation, we have $5,533$ parameters.

On $\mathbb{S}^3$, we consider an unnormalized density proportional to

$$\pi_{\mathbb{S}^3}(y) \propto \sum_{i=1}^{4} \exp(10 y^\top \mu_i) \tag{43}$$

$$\mu_1 = (-0.129, 0.070, 0.659, -0.738) \tag{44}$$

$$\mu_2 = (-0.990, -0.076, 0.118, -0.017) \tag{45}$$

$$\mu_3 = (0.825, -0.484, 0.061, 0.285) \tag{46}$$

$$\mu_4 = (-0.801, 0.592, -0.024, 0.081). \tag{47}$$

When using RealNVP, the total number of learnable parameters in our dequantization model is $21,854$; dequantization with an ambient ODE has $21,386$ learnable parameters; in the case of the Möbius transform the total number of learnable parameters is $25,406$; for the manifold ODE implementation, we have $21,204$ parameters. For dequantization we use 100 samples per batch and use rejection sampling to draw samples from the unnormalized target density at each iteration.

## E.2 Torus

Expressed in terms of their *angular* coordinates (as opposed to their embedding into $\mathbb{R}^4$), the densities on the torus are as follows:

**Unimodal** $\pi_{\mathbb{T}^2}^{\mathrm{uni}}(\theta_1, \theta_2 | \phi) \propto \exp(\cos(\theta_1 - \phi_1) + \cos(\theta_2 - \phi_2))$ with $\phi = (4.18, 5.96)$.

**Multimodal** $\pi_{\mathbb{T}^2}^{\mathrm{mul}}(\theta_1, \theta_2) \propto \sum_{i=1}^{3} \pi_{\mathbb{T}^2}^{\mathrm{uni}}(\theta_1, \theta_2 | \phi_i)$ where $\phi_1 = (0.21, 2.85)$, $\phi_2 = (1.89, 6.18)$, and $\phi_3 = (3.77, 1.56)$.

**Correlated** $\pi_{\mathbb{T}^2}^{\mathrm{cor}}(\theta_1, \theta_2) \propto \exp(\cos(\theta_1 + \theta_2 - 1.94))$.

The number of learnable parameters in the dequantization models is $6,106$; in the Möbius flow model, the number of learnable parameters is $5,540$; for the direct method, the number of learnable parameters is $5,406$. For dequantization we use 100 samples per batch and use rejection sampling to draw samples from the unnormalized target density at each iteration.

## E.3 Orthogonal Group

Drawing inspiration from the orthogonal Procrustes problem, we define the unnormalized density by

$$\pi_{\mathrm{O}(n)}(\mathbf{O}) \propto \exp\left(-\frac{1}{2\sigma^2} \|\mathbf{B} - \mathbf{O}\mathbf{A}\|_{\mathrm{fro}}^2\right), \tag{48}$$

where $\mathbf{A}, \mathbf{B} \in \mathbb{R}^{n \times p}$. Given samples from eq. (48), we may apply our dequantization procedure to perform density estimation. In our experiments we take $p = 10$ and $n = 3$. For dequantization we use rejection sampling to draw samples from the posterior. Then, with these fixed samples, we use batches of 100 samples to train the ambient and dequantization distributions.

### E.4 Special Orthogonal Group

Let $\mathbf{R} \in \mathbb{R}^{n \times n}$ be a reflection matrix and notice that $\{\mathrm{SO}(n), \mathbf{R}\mathrm{SO}(n)\}$ is partition of $\mathrm{O}(n)$. Given a density on $\mathbb{R}^{n \times n}$, using the methods described in section 5.1.3, we may obtain a density on $\mathrm{O}(n)$. Then, we define the function $S : \mathrm{O}(n) \to \mathrm{SO}(n)$ by

$$S(\mathbf{O}) \overset{\text{def.}}{=} \begin{cases} \mathbf{O} & \text{if } \det(\mathbf{O}) = +1 \\ \mathbf{R}\mathbf{O} & \text{if } \det(\mathbf{O}) = -1 \end{cases}, \tag{49}$$

where $\mathbf{R}$ is a reflection matrix. Now, define $S_1(\mathbf{O}) \overset{\text{def.}}{=} \mathbf{O}$ and $S_2(\mathbf{O}) \overset{\text{def.}}{=} \mathbf{R}\mathbf{O}$, which satisfy $S_1 = S|\mathrm{SO}(n)$ and $S_2 = S|\mathbf{R}\mathrm{SO}(n)$. Both $S_1$ and $S_2$ are self-inverse and volume-preserving maps on their respective domains so we may obtain a density on $\mathrm{SO}(n)$ as

$$\pi_{\mathrm{SO}(n)}(\mathbf{O}) = \pi_{\mathrm{O}(n)}(\mathbf{O}) + \pi_{\mathrm{O}(n)}(\mathbf{R}\mathbf{O}). \tag{50}$$

One may immediately seek to minimize the negative log-likelihood of data using eq. (50). Alternatively, we use the following ELBO in our our experiments:

$$\log \pi_{\mathrm{SO}(n)}(\mathbf{O}) \geq \underset{\mathbf{S} \sim \mathrm{Unif}(\mathrm{Id}_n, \mathbf{R})}{\mathbb{E}} \underset{\mathbf{L} \sim \tilde{\pi}_{\mathrm{Tri}_+}}{\mathbb{E}} \frac{\pi_{\mathbb{R}^{n \times n}}(\mathbf{SOP})}{\det(\nabla G(\mathbf{SOP})) \cdot \tilde{\pi}_{\mathrm{Tri}_+}(\mathbf{L})}, \tag{51}$$

where, as in section 5.2.3, $\mathbf{P} = \mathbf{L}\mathbf{L}^\top$. For a short derivation of this ELBO, see appendix A.

We consider the following multimodal density on $\mathrm{SO}(3)$:

$$\pi_{\mathrm{SO}(3)}(\mathbf{O}) \propto \sum_{i=1}^{3} \exp(-\frac{1}{2\sigma^2} \|\mathbf{O} - \Omega_i\|_{\mathrm{fro}}^2) \tag{52}$$

where in our experiments we set $\sigma = 1/2$, $\Omega_1 = \mathrm{diag}(1, 1, 1)$, $\Omega_2 = \mathrm{diag}(-1, -1, 1)$, and $\Omega_3 = \mathrm{diag}(-1, 1, -1)$. For dequantization we use rejection sampling to draw samples from the posterior. Then, with these fixed samples, we use batches of 100 samples to train the ambient and dequantization distributions.

# F   Practical Considerations

| Element | Notation | Description |
|---|---|---|
| Ambient Euclidean Space | $\mathcal{X} = \mathbb{R}^m$ | Euclidean space in which the manifold of interest is embedded |
| Ambient Density | $\pi_{\mathcal{X}}(\cdot\|\theta)$ | A flexible family of densities parameterized by $\theta \in \mathbb{R}^{n_{\text{amb}}}$ on $\mathcal{X}$ |
| Change-of-Variables Mapping | $G : \mathcal{X} \to \mathcal{Y} \times \mathcal{Z}$ | A smooth mappingsatisfying the conditions of corollary 1 or theorem 1 where $\mathcal{Z}$ is an auxiliary manifold. |
| Dequantization Density | $\tilde{\pi}_{\mathcal{Z}}(\cdot\|\phi, y)$ | A non-vanishing family of dequantization distributions parameterized by $\phi \in \mathbb{R}^{n_{\text{deq}}}$ and possibly depending on $y \in \mathcal{Y}$. |
| Loss Function | $\mathcal{L} : \mathbb{R}^{n_{\text{amb}} \times n_{\text{deq}}} \to \mathbb{R}$ | A loss function depending on $\left\{ y_1, \ldots, y_{n_{\text{obs}}} \right\}$, $\pi_{\mathcal{X}}$, and $\tilde{\pi}_{\mathcal{Z}}$ that is differentiable in $\theta$ and $\phi$ and captures the quality of the density estimate. |

Table 5: The five elements that we use to dequantize a manifold into an ambient Euclidean space using a change-of-variables and dequantization density for marginalization.

## F.1   The Ambient Euclidean Space

It is frequently the case that a manifold has a natural embedding into Euclidean space. For instance, the sphere $\mathbb{S}^{m-1}$ is naturally embedded into $\mathbb{R}^m$. The Stiefel manifold is a subset of $\mathbb{R}^{n \times p}$ satisfying an orthonormality condition; therefore, it is naturally embedded into $\mathbb{R}^{n \times p}$. For some manifolds, the choice may require some consideration. For instance, a torus $\mathbb{T}^2$ may be regarded as a product manifold of two circles, each of which are naturally embedded into $\mathbb{R}^2$ so that the entire torus is embedded into $\mathbb{R}^4$ (this is called the *Clifford torus*). An alternative is the familiar embedding of the torus as a "doughnut" in $\mathbb{R}^3$, although in this case it may be non-obvious how to construct a suitable mapping $G$ between $\mathbb{R}^3$, the doughnut torus, and some auxiliary choice of one-dimensional manifold.

## F.2   The Change-of-Variables and Auxiliary Manifold

We have seen several examples wherein manifolds of interest appear alongside an auxiliary manifold when an ambient Euclidean space is transformed under a change-of-variables. In general, one would like to identify a transformation $G : \mathcal{X} \to \mathcal{Y} \times \mathcal{Z}$, satisfying the conditions of theorem 1 or corollary 1, such that it will be straightforward to formulate importance sampling random variables on $\mathcal{Z}$ so as to obtain the marginal density on $\mathcal{Y}$.

## F.3   Dequantization Density

Dequantization densities $\tilde{\pi}_{\mathcal{Z}}$ on $\mathcal{Z}$ has to respect the constraints of the auxiliary manifold. For instance, if $\mathcal{Z} = \mathbb{R}_+$ we must choose a dequantization density whose support is the positive real numbers and which is non-vanishing. The requirement that $\tilde{\pi}_{\mathcal{Z}}$ be non-vanishing (that is, $\tilde{\pi}_{\mathcal{Z}}(z) > 0$ for all $z \in \mathcal{Z}$) is important so as to avoid division-by-zero singularities in the importance sampling formula eq. (10). We have already seen some examples of dequantization densities that respect the constraints of the auxiliary manifold in section 5. In each of these cases, the dequantization distribution is parameterized (for example, the Gamma distribution is parameterized by its shape and scale); it may be desirable to choose these parameters to depend on the location $y \in \mathcal{Y}$. To accomplish this, one may construct a neural network with parameters $\phi$ and input $y$ whose output is the parameterization of the dequantization distribution. Thus the dequantization distribution may be in general expressed as $\tilde{\phi}_{\mathcal{Z}}(\cdot\|\phi, y)$.

## G   Evaluation Metrics

Given a density $\pi_{\mathcal{Y}}(y) \propto \exp(-u(y))$, known up to proportionality, we consider several performance metrics for the dequantization method we propose. The error of the first order moment is defined by,

$$\| \mathop{\mathbb{E}}_{y \sim \pi_{\mathcal{Y}}} [y] - \mathop{\mathbb{E}}_{\hat{y} \sim \hat{\pi}_{\mathcal{Y}}} [\hat{y}] \|_2. \tag{53}$$

The error of the second (centered) moment is defined by

$$\| \mathop{\mathrm{Cov}}_{y \sim \pi_{\mathcal{Y}}} (y) - \mathop{\mathrm{Cov}}_{\hat{y} \sim \hat{\pi}_{\mathcal{Y}}} (\hat{y}) \|_{\mathrm{fro}}. \tag{54}$$

The Kullback-Leibler divergence of $\hat{\pi}_{\mathcal{Y}}$ and $\pi_{\mathcal{Y}}$ is

$$\mathrm{KL}(\hat{\pi}_{\mathcal{Y}} \| \pi_{\mathcal{Y}}) \overset{\mathrm{def.}}{=} \mathop{\mathbb{E}}_{\hat{y} \sim \hat{\pi}_{\mathcal{Y}}} \log \frac{\hat{\pi}_{\mathcal{Y}}(\hat{y})}{\pi_{\mathcal{Y}}(\hat{y})} \tag{55}$$

$$= \mathop{\mathbb{E}}_{\hat{y} \sim \hat{\pi}_{\mathcal{Y}}} [\log \hat{\pi}_{\mathcal{Y}}(\hat{y}) + u(\hat{y})] + \log Z, \tag{56}$$

where $Z$ is the normalizing constant of $\pi_{\mathcal{Y}}(y)$. We can estimate $Z$ via importance sampling according to

$$Z = \mathop{\mathbb{E}}_{\hat{y} \sim \hat{\pi}_{\mathcal{Y}}} \left[ \frac{\exp(-u(\hat{y})))}{\hat{\pi}_{\mathcal{Y}}(\hat{y})} \right], \tag{57}$$

which permits us to compute a Monte Carlo estimate of the Kullback-Leibler divergence. The reverse direction of the KL-divergence $\mathrm{KL}(\pi_{\mathcal{Y}} \| \hat{\pi}_{\mathcal{Y}})$ may be similarly computed.

Let $\{\hat{y}_1, \ldots, \hat{y}_n\}$ be a collection of independent identically-distributed samples from $\hat{\pi}_{\mathcal{Y}}$. The number of effective independent samples is the quantity,

$$\mathrm{ESS} \overset{\mathrm{def.}}{=} \frac{\left( \sum_{i=1}^{n} \omega_i \right)^2}{\sum_{i=1}^{n} \omega_i^2} \tag{58}$$

where $\omega_i = \exp(-u(\hat{y}_i))/\hat{\pi}_{\mathcal{Y}}(\hat{y}_i)$. See (Liu, 2008; Gower & Dijksterhuis, 2004) for details on the ESS metric. Following (Rezende et al., 2020), we report the relative ESS, which the ratio of the effective sample size and $n$, the number of samples. When a Monte Carlo approximation of the evaluation metric is required, we use rejection sampling in order to obtain samples from the density $\pi_{\mathcal{Y}}$.

# H  Proof of Manifold Change-of-Variables Formula

**Proposition 4.** Let $A \subseteq \mathbb{R}^m$ and let $G : A \to \mathbb{R}^n$ be a smooth function. Suppose $m < n$. Define the embedded manifold $M \stackrel{\text{def.}}{=} G(A)$. Let $\pi$ be a real-valued, continuous function on $M$. Then,

$$\int_M \pi(x) \, \mathrm{dVol}(x) = \int_A \pi(G(t)) \cdot \sqrt{\det((\nabla G(t))^\top (\nabla G(t)))} \, \mathrm{d}t \tag{59}$$

where dVol is the volume measure on $M$.

*Proof.* See (Jones, 2004). □

**Corollary 3.** Define,

$$\pi_{\mathbb{R}^m}(t) \stackrel{\text{def.}}{=} \pi(G(t)) \cdot \sqrt{\det((\nabla G(t))^\top (\nabla G(t)))}. \tag{60}$$

Then rearranging immediately implies,

$$\pi(G(t)) = \frac{\pi_{\mathbb{R}^m}(t)}{\sqrt{\det((\nabla G(t))^\top (\nabla G(t)))}}, \tag{61}$$

which is the manifold change-of-variables formula.

What follows now is a more detailed exposition on this result using Riemannian geometry.

Let $U$ be an open subset of $\mathbb{R}^m$. Let $G$ be a continuous function from $U \to \mathbb{R}^n$ that is a homeomorphism on its image. That is, defining $\mathcal{M} \stackrel{\text{def.}}{=} \{G(x) : x \in U\} \subset \mathbb{R}^n$, we find that $G$ has a continuous inverse on $\mathcal{M}$. As a subset of $\mathbb{R}^n$, we can equip $\mathcal{M}$ with the subspace topology and find that $\mathcal{M}$ is a topological $m$-manifold. Under these assumptions, it immediately follows that $(\mathcal{M}, G^{-1})$ is a *global* coordinate chart in the sense of differential geometry.

If we further assume that $G$ and its inverse are smooth functions (in the sense of ordinary calculus), then it follows that $\mathcal{M}$ is globally diffeomorphic to $U$. Evidently, our discussion allows $m \leq n$. If the Jacobian (in the sense of ordinary calculus) of $G$, denoted $\nabla G : U \to \mathbb{R}^{n \times m}$, has the property that at every $x \in U$, $\mathrm{rank}(\nabla G(x)) = m$, then $G$ is called a smooth immersion; this is equivalent to $\nabla G$ having full-rank at each $x \in U$. Moreover, because $G$ is homeomorphic onto its image, $G$ is also a smooth embedding.

The tangent space of a smooth manifold can be constructed as the vector space of velocities that a particle moving along the manifold may exhibit at a point. Formally, let $(a, b) \subset \mathbb{R}$ be an open interval and let $x : (a, b) \to U$ be a parameterized smooth curve in $U$. The composition $y \stackrel{\text{def.}}{=} G \circ x : (a, b) \to \mathcal{M}$ is then a parameterized smooth curve on $\mathcal{M}$. The velocity of $y$ is then computed as,

$$\frac{\mathrm{d}}{\mathrm{d}t} y(t) = \frac{\mathrm{d}}{\mathrm{d}t}(G \circ x)(t) = \nabla G(x(t)) \dot{x}(t). \tag{62}$$

From the preceding discussion, $\nabla G(x(t)) \in \mathbb{R}^{n \times m}$ is a matrix of full-rank and therefore has $m$ linearly-independent $n$-dimensional columns. Since $\dot{x}(t) \in \mathbb{R}^m$ can be arbitrary, we find that the tangent space of $\mathcal{M}$ at $y(t)$ is the vector space spanned by the columns of $\nabla G(x(t))$; note that these basis vectors depend only on the position in $U$ and not on time. Given $y \in \mathcal{M}$, we denote the tangent space by $\mathrm{T}_y \mathcal{M} = \{\nabla G(x)c : c \in \mathbb{R}^m, x = G^{-1}(y)\}$ which is a vector subspace of $\mathbb{R}^n$. In the following, we make use of the common identification $\mathrm{T}_x \mathbb{R}^m \cong \mathbb{R}^m$.

A smooth manifold can be turned into a Riemannian manifold by equipping its tangent spaces with an inner product called the Riemannian metric. Formally, given $y \in \mathcal{M}$, $g(y) \mapsto \langle \cdot, \cdot \rangle_y$ where $\langle \cdot, \cdot \rangle_y : \mathrm{T}_y \mathcal{M} \times \mathrm{T}_y \mathcal{M} \to \mathbb{R}$ is an inner product. Given the discussion so far, there is no prescription for a Riemannian metric. However, we may "prefer" the Riemannian metric that is induced from the ambient Euclidean space $\mathbb{R}^n$. For $\tilde{y} \in \mathbb{R}^n$, the Euclidean metric is defined by $\langle u, v \rangle_{\tilde{y}} = u^\top v$ where $u, v \in \mathrm{T}_{\tilde{y}} \mathbb{R}^n \cong \mathbb{R}^n$. The induced metric on $\mathcal{M} \subset \mathbb{R}^n$ is then defined by $\langle u, v \rangle_y = u^\top v$ where $u$ and $v$ are viewed as vectors in $\mathbb{R}^n$ satisfying $u, v \in \mathrm{T}_y \mathcal{M}$.

Given a metric on $\mathcal{M}$, we can compute an associated metric on $U$ called the *pullback metric*. The pullback metric is defined by,

$$\tilde{g}(x)(\tilde{u}, \tilde{v}) = g(G(x))(\nabla G(x)\tilde{u}, \nabla G(x)\tilde{v}) \tag{63}$$

$$= \tilde{u}^\top \nabla G(x)^\top \nabla G(x)\tilde{v} \tag{64}$$

where $\tilde{u}, \tilde{v} \in \mathrm{T}_x\mathbb{R}^m \cong \mathbb{R}^m$. From the condition that $\nabla G(x)$ is a matrix of full-rank for each $x \in U$, it can be shown that $\nabla G(x)^\top \nabla G(x)$ is a positive definite matrix. Hence the pullback metric is a proper inner product. Unlike the induced metric on $\mathcal{M}$ which has no real dependency on $y \in \mathcal{M}$, the pullback metric does depend on $x \in U$ through $\nabla G(x)^\top \nabla G(x)$. The pullback metric can be regarded as the expression of the induced metric on $\mathcal{M}$ in the global coordinates of $U$. Indeed, if $y = G(x)$, recall that every $u \in \mathrm{T}_y\mathcal{M}$ can be expressed as $u = \nabla G(x)\tilde{u}$ where $\tilde{u} \in \mathbb{R}^m$; the unique $\tilde{u}$ can be computed from

$$\tilde{u} = (\nabla G(x)^\top \nabla G(x))^{-1} \nabla G(x)^\top u. \tag{65}$$

In our construction, we have described how to induce a metric on $\mathcal{M}$ via the Euclidean metric in the embedding space. We then described how the metric materializes in the global coordinate system via the pullback metric. We are now in a position to state how these objects inform integration on Riemannian manifolds. We have the following theorem.

**Theorem 2.** Let dVol denote the Riemannian volume element on $\mathcal{M}$ when the metric on $\mathcal{M}$ is the induced Euclidean metric from $\mathbb{R}^n$. Given a smooth function $f : \mathcal{M} \to \mathbb{R}$, its integral over $\mathcal{M}$ can be expressed in terms of the global coordinate system as,

$$\int_{\mathcal{M}} f(y) \, \mathrm{dVol}(y) = \int_U f(G(x)) \cdot \sqrt{\det(\nabla G(x)^\top \nabla G(x))} \, \mathrm{d}x. \tag{66}$$

A proof of this result may be found in (Jones, 2004). More sophisticated variants of this result may be found textbooks on differential geometry; see, *inter alia*, (Lee, 2003).

If $\pi_{\mathcal{M}}$ is a density on $\mathcal{M}$, $y \sim \pi_{\mathcal{M}}$ and $A \subset \mathcal{M}$, then we have,

$$\Pr_{y \sim \pi_{\mathcal{M}}} [y \in A] = \int_{\mathcal{M}} \mathbf{1}\{y \in A\} \cdot \pi_{\mathcal{M}}(y) \, \mathrm{dVol}(y) \tag{67}$$

$$= \int_U \pi_{\mathcal{M}}(G(x)) \cdot \mathbf{1}\{G(x) \in A\} \cdot \sqrt{\det(\nabla G(x)^\top \nabla G(x))} \, \mathrm{d}x \tag{68}$$

$$= \Pr_{x \sim \pi_U} [x \in G^{-1}(A)] \tag{69}$$

where

$$\pi_U(x) = \pi_{\mathcal{M}}(G(x)) \cdot \sqrt{\det(\nabla G(x)^\top \nabla G(x))} \tag{70}$$

which, upon rearrangement, is the manifold change-of-variables formula.

Given a subset $A \subset \mathcal{M}$, we define its volume by,

$$\int_A \mathrm{dVol}(y) = \int_{G^{-1}(A)} \sqrt{\det(\nabla G(x)^\top \nabla G(x))} \, \mathrm{d}x. \tag{71}$$

Such a definition of volume is sometimes called the "surface measure" and it is a generalization of arclength in the one-dimensional setting. Moreover, this notion of volume coincides with the $m$-dimensional Hausdorff area of $A$; this is a consequence of the *area formula* and a detailed discussion can be found in (Federer, 1969).

# I Embedded Manifolds

A smooth manifold $\mathcal{X}$ of dimension $k$ is a second-countable Hausdorff space such that for every $x \in \mathcal{X}$ there is a homeomorphism between a neighborhood of $x$ and $\mathbb{R}^k$. By the Whitney Embedding Theorem (see, inter alia, (Lee, 2003)), every smooth manifold can be smoothly embedded into Euclidean space of dimension $2k$. It is frequently possible to express an embedded manifold as the zero level-set of a constraint function: Let $g : \mathbb{R}^m \to \mathbb{R}^k$ be a smooth function and define $\mathcal{X} \stackrel{\text{def.}}{=} \{x \in \mathbb{R}^m : g(x) = 0\}$. If $\nabla g(x) \in \mathbb{R}^{k \times m}$ is a matrix of full-rank for every $x \in \mathcal{X}$, we say that $\mathcal{X}$ is an embedded manifold of rank $k$. To see how these definitions apply, let us consider some examples.

## I.1 Hypersphere

The sphere in $\mathbb{R}^3$ is the zero level-set of the constraint function $g(x) \stackrel{\text{def.}}{=} x^\top x - 1$.

## I.2 Torus

The torus is the preimage of the constraint function $g : \mathbb{R}^4 \to \mathbb{R}^2$ defined by

$$g(x) \stackrel{\text{def.}}{=} \begin{pmatrix} x_1^2 + x_2^2 - 1 \\ x_3^2 + x_4^2 - 1 \end{pmatrix}. \tag{72}$$

## I.3 Stiefel Manifold

A constraint function for the Stiefel$(n, p)$ manifold is $g : \mathbb{R}^{n \times p} \to \mathbb{R}^{n \times n}$ defined by $g(\mathbf{M}) = \mathbf{M}^\top \mathbf{M} - \text{Id}_n$. Using the vec $: \mathbb{R}^{m \times n} \to \mathbb{R}^{mn}$ isomorphism, Stiefel$(n, p)$ may also be embedded into $\mathbb{R}^{np}$.

## J   Stiefel QR Decomposition

In appendix D we discussed computing the positive definite component $\mathbf{P}$ as the principal square root of $\mathbf{A}^\top \mathbf{A}$. In our algorithm, $\mathbf{P}$ is represented according to its Cholesky factor, which is unique because $\mathbf{P}$ is (assuming $\mathbf{A}$ is full-rank). An alternative would be to compute a Cholesky factor of $\mathbf{A}^\top \mathbf{A}$ directly; such an approach allows us to use the QR decomposition in place of the polar decomposition.

**Proposition 5.** Suppose $\mathbf{A} \in \mathbb{R}^{m \times n}$ is a matrix of full-rank. The (unique) QR decomposition of $\mathbf{A}$ is

$$\mathbf{A} = \mathbf{QR} \tag{73}$$

where

$$\mathbf{L} \overset{\text{def.}}{=} \text{Cholesky}(\mathbf{A}^\top \mathbf{A}) \tag{74}$$

$$\mathbf{R} \overset{\text{def.}}{=} \mathbf{L}^\top \tag{75}$$

$$\mathbf{Q} \overset{\text{def.}}{=} \mathbf{A}\mathbf{R}^{-1}. \tag{76}$$

Moreover $\mathbf{Q} \in \text{Stiefel}(m, n)$.

*Proof.* The uniqueness of $\mathbf{L}$ (and consequently $\mathbf{R}$) follows from the fact that the Cholesky decomposition of positive definite matrices is unique. Since $\mathbf{A}$ is of full-rank, $\mathbf{A}^\top \mathbf{A}$ is a positive definite matrix. Since $\mathbf{L}$ has positive diagonal entries, its determinant is non-zero and therefore has a unique inverse. This implies the uniqueness of $\mathbf{Q}$. The fact that $\mathbf{Q}$ is an element of the Stiefel manifold follows from direct evaluation:

$$\mathbf{Q}^\top \mathbf{Q} = (\mathbf{R}^{-1})^\top \mathbf{A}^\top \mathbf{A} \mathbf{R}^{-1} \tag{77}$$

$$= \mathbf{L}^{-1} \mathbf{L} \mathbf{L}^\top (\mathbf{L}^\top)^{-1} \tag{78}$$

$$= \text{Id.} \tag{79}$$

$\square$

**Proposition 6.** Let $(\mathbf{Q}, \mathbf{L}) \in \text{Stiefel}(m, n) \times \text{Tri}_+(n)$. The Jacobian determinant of the transformation $(\mathbf{Q}, \mathbf{L}) \mapsto \mathbf{Q}\mathbf{L}^\top$ is

$$\prod_{i=1}^{n} \mathbf{L}_{ii}^{m-i}. \tag{80}$$

*Proof.* See page 31 of (Edelman, 1989). $\square$

# K   Overview of Normalizing Flows

This section describes the Euclidean normalizing flows used in this paper.

## K.1   RealNVP

Let $x \in \mathbb{R}^m$ and let $n, p \in \mathbb{N}$ satisfy $m = n + p$. Consider partitioning $x$ into components $x_a \in \mathbb{R}^n$ and $x_b \in \mathbb{R}^p$ by taking the first $n$ and last $p$ components from $x$, respectively. Let $\mu(\cdot; \theta) : \mathbb{R}^n \to \mathbb{R}^p$ and $\sigma : \mathbb{R}^n \to \mathbb{R}_+^p$ be a functions parameterized by $\theta \in \mathbb{R}^{d_\mu}$ and $\phi \in \mathbb{R}^{d_\sigma}$, respectively. Compute the affine transformation of $x_b$ according to

$$y_b \stackrel{\text{def.}}{=} \sigma(x_a) \odot x_b + \mu(x_a) \tag{81}$$

$$y \stackrel{\text{def.}}{=} (x_a, y_b). \tag{82}$$

The transformation $x \mapsto y$ has a Jacobian of the form,

$$\nabla_x y = \begin{pmatrix} \text{Id}_n & \mathbf{0}_{n \times p} \\ \nabla_{x_a} y_b & \text{diag}(\sigma(x_a)) \end{pmatrix}. \tag{83}$$

The key observation about this Jacobian is that it has a lower-triangular structure. Therefore, its determinant is the product of its diagonal elements.

The transformation $x \mapsto y$ is also invertible. Let $(y_a, y_b)$ be the corresponding partition of $y$. The inverse map is given by,

$$x_b = (y_b - \mu(y_a)) \oslash \sigma(y_a) \tag{84}$$

$$x_a = y_a. \tag{85}$$

With the inverse and Jacobian determinant available, we may apply the transformation $x \mapsto y$ in the Euclidean change-of-variables formula. This transformation is called RealNVP. In practice, it is common to chain multiple RealNVP transformations together in order to obtain a more expressive flow. One may also permute the elements of $y$ after each application of the RealNVP transformation (which is, of course, an invertible, volume-preserving transformation) in order to construct affine transformations of different variables.

## K.2   Neural ODE

Let $x \in \mathbb{R}^m$ and let $f(\cdot, \cdot; \theta) : \mathbb{R}^m \times \mathbb{R} \to \mathbb{R}^m$ be a smooth function parameterized by $\theta \in \mathbb{R}^d$. Consider solving the initial value problem defined by,

$$\frac{\mathrm{d}}{\mathrm{d}t} \phi_t(x) = f(\phi_t(x), t; \theta) \tag{86}$$

$$\phi_0(x) = x. \tag{87}$$

The map $\phi_{(\cdot)}(\cdot) : \mathbb{R} \times \mathbb{R}_m \to \mathbb{R}_m$ is called the flow of $f$. The existence and uniqueness of differential equations leads to the group property of flows:

$$\phi_{t+s}(x) = \phi_t(\phi_s(x)). \tag{88}$$

In particular, since $\phi_0(x) = x$, we have $\phi_{-t} \circ \phi_t = \text{Id}$ or $\phi_t^{-1} = \phi_{-t}$ so that the inverse flow map is obtained from the flow map with a negated time index.

How does the flow of a vector field affect probability? This is an important question that can be answered at varying levels of sophistication. One elegant analysis uses techniques from fluid mechanics and the Lie

derivative theorem. In the following we adopt a simpler, if more mechanical, derivation of the rate of change of the Jacobian determinant.

$$\frac{\mathrm{d}}{\mathrm{d}t}\det(\nabla_x\phi_t(x)) = \det(\nabla_x\phi_t(x))\ \mathrm{trace}\left([\nabla_x\phi_t(x)]^{-1}\frac{\mathrm{d}}{\mathrm{d}t}\nabla_x\phi_t(x)\right) \tag{89}$$

$$= \det(\nabla_x\phi_t(x))\ \mathrm{trace}\left([\nabla_x\phi_t(x)]^{-1}\mathbf{D}_x f(\phi_t(x), t; \theta)\right) \tag{90}$$

$$= \det(\nabla_x\phi_t(x))\ \mathrm{trace}\left([\nabla_x\phi_t(x)]^{-1}\nabla_x f(\phi_t(x), t; \theta)\nabla_x\phi_t(x)\right) \tag{91}$$

$$= \det(\nabla_x\phi_t(x))\ \mathrm{trace}\left(\nabla_x f(\phi_t(x), t; \theta)\right) \tag{92}$$

$$= \det(\nabla_x\phi_t(x))\ \mathrm{div}(f(\phi_t(x), t; \theta)) \tag{93}$$

From which it follows that,

$$\frac{\mathrm{d}}{\mathrm{d}t}\log|\det(\nabla_x\phi_t(x))| = \mathrm{div}(f(\phi_t(x), t; \theta)). \tag{94}$$

Let $\pi : \mathbb{R}^m \to \mathbb{R}_+$ be a probability density and let $x \sim \pi$. Let $\pi_t(\phi_t(x))$ be the density of $\phi_t(x)$. Because $\log\pi_t(\phi_t(x)) = \log\pi(x) - \log|\det(\nabla_x\phi_t(x))|$, we have,

$$\frac{\mathrm{d}}{\mathrm{d}t}\log\pi_t(\phi_t(x)) = -\mathrm{div}(f(\phi_t(x), t; \theta)). \tag{95}$$

This formula is sometimes called the *instantaneous change-of-variables formula*. From the initial condition $\phi_0(x) = x$, we obtain that $\log|\det(\nabla_x\phi_0(x))| = 0$, which gives an initial condition for the time evolution of the Jacobian determinant.

In general, none of the initial value problems described so far have analytical solutions. This necessitates the use of numerical integrators to compute the map $\phi_t(\cdot)$, its inverse, and the Jacobian determinant correction from the instantaneous change-of-variables formula. The method is called *neural ODE* because $f(\cdot, \cdot; \theta)$ is often chosen to be a neural network parameterized by $\theta$.

## L    Universal Approximation for the Dequantized Density

The purpose of this section is to give sufficient conditions on the dequantization of a distribution on a manifold into an ambient Euclidean space such that a suitably expressive normalizing flow on the Euclidean space could, in principle, learn the density to an arbitrary precision. The conclusion of this section is that it is sufficient that the distribution on the manifold and auxiliary structure be measurable with respect to the natural Borel $\sigma$-algebra and that the transformation between Euclidean space and the product space of the manifold and auxiliary structure be continuously differentiable. After stating this result, the remainder of the section is devoted to explaining what is meant by universal approximation of Euclidean densities.

**Definition 4.** Let $\mathcal{X}$ be a topological space. The smallest $\sigma$-algebra containing all the open sets of $\mathcal{X}$ is called the Borel $\sigma$-algebra. The Borel $\sigma$-algebra is denoted $\mathfrak{B}(\mathcal{X})$. An element $A \subset \mathfrak{B}(\mathcal{X})$ is called a Borel set.

**Definition 5.** A measure $\mu$ on $\mathbb{R}^m$ is said to be absolutely continuous with respect to the Lebesgue measure if there exists a measurable function $\pi : \mathbb{R}^m \to \mathbb{R}_+$ such that

$$\mu(A) = \int_A \pi(x) \, \mathrm{d}x, \tag{96}$$

where $A \in \mathfrak{B}(\mathbb{R}^m)$.

**Definition 6.** Let $\mathcal{X}$ and $\mathcal{Y}$ be topological spaces. A function $f : \mathcal{X} \to \mathcal{Y}$ is said to be Borel measurable if for all $E \in \mathfrak{B}(\mathcal{Y})$

$$f^{-1}(E) = \{x \in \mathcal{X} : f(x) \in E\} \in \mathfrak{B}(\mathcal{X}). \tag{97}$$

**Proposition 7.** Let $f : \mathcal{X} \to \mathcal{Y}$ and $g : \mathcal{X} \to \mathcal{Y}$ be two Borel measurable functions. Then the product function $h(x) \stackrel{\text{def.}}{=} g(x)f(x)$ is also Borel measurable. Moreover, if $k : \mathcal{Y} \to \mathcal{Z}$ is another Borel measurable function then the composition $k \circ f : \mathcal{X} \to \mathcal{Z}$ is also Borel measurable.

**Proposition 8.** Let $U$ be an open subset of $\mathbb{R}^m$ and let $G : U \to \mathbb{R}^n$ be a homeomorphism on its image. Let $\mathcal{M} \stackrel{\text{def.}}{=} \{G(x) : x \in U\}$. Assume further that $G$ is a continuously differentiable function. Let $\mu_{\mathcal{M}}$ be a measure on $\mathcal{M}$ that is absolutely continuous with respect to the volume measure on $\mathcal{M}$; that is,

$$\mu_{\mathcal{M}}(A) = \int_A \pi_{\mathcal{M}}(y) \, \mathrm{dVol}(y) \tag{98}$$

where $\pi_{\mathcal{M}} : \mathcal{M} \to \mathbb{R}_+$ is a measurable function and $A \subset \mathcal{M}$ is a Borel subset. Then, if $y \sim \pi_{\mathcal{M}}$, the random variable $x \stackrel{\text{def.}}{=} G^{-1}(y)$ has a measurable density with respect the the Lebesgue measure given by

$$\pi_U(x) \stackrel{\text{def.}}{=} \pi_{\mathcal{M}}(G(x)) \cdot \sqrt{\det(\nabla G(x)^\top \nabla G(x))}. \tag{99}$$

*Proof.* By the manifold change-of-variables formula, the measure $\mu_{\mathcal{M}}$ is related to the Lebesgue measure on $\mathbb{R}^m$ according to,

$$\mu_{\mathcal{M}}(A) = \int_{G^{-1}(A)} \pi_{\mathcal{M}}(G(x)) \cdot \sqrt{\det(\nabla G(x)^\top \nabla G(x))} \, \mathrm{d}x. \tag{100}$$

Now we apply the results of proposition 7 to show that the integrand on the right-hand side of eq. (100) is a measurable function. Because $G$ is smooth it is continuous, and continuous functions are measurable. Therefore, when $A \in \mathfrak{B}(\mathcal{M})$ we obtain that $G^{-1}(A) \in \mathfrak{B}(U)$. Moreover, $\pi_{\mathcal{M}} \circ G$ is a measurable function with respect to the Borel $\sigma$-algebra on $U$ because the composition of measurable functions is measurable. Finally, since $G$ is continuously differentiable, and since the determinant and square-root are continuous functions, $\sqrt{\det(\nabla G(x)^\top \nabla G(x))}$ is also a measurable function.

Thus, if $y \sim \pi_{\mathcal{M}}$, the random variable $x \stackrel{\text{def.}}{=} G^{-1}(y)$ has a measurable density with respect the the Lebesgue measure given by

$$\pi_U(x) \stackrel{\text{def.}}{=} \pi_{\mathcal{M}}(G(x)) \cdot \sqrt{\det(\nabla G(x)^\top \nabla G(x))}. \tag{101}$$

$\square$

**Corollary 4.** The measure

$$\mu_U(A) \stackrel{\text{def.}}{=} \int_A \pi_U(x) \, \mathrm{d}x \tag{102}$$

is absolutely continuous with respect to the Lebesgue measure.

In the context of dequantization, we would like to understand the conditions under which the dequantization of a density on an embedded manifold $\mathcal{Y}$ via an auxiliary embedded manifold $\mathcal{Z}$ and a smooth, invertible function $G : U \to \mathcal{Y} \times \mathcal{Z}$ produces a Lebesgue measurable density on $U \subset \mathbb{R}^m$.

**Proposition 9.** Let $\mathcal{Y}$ and $\mathcal{Z}$ be embedded manifolds. Let $\pi_{\mathcal{Y}}$ be the density on $\mathcal{Y}$ defined with respect to the Riemannian volume element $\mathrm{dVol}_{\mathcal{Y}}$ and let $\pi_{\mathcal{Z}}(\cdot|y)$ be a density on $\mathcal{Z}$ (which may depend on $y \in \mathcal{Y}$) defined with respect to the Riemannian volume element $\mathrm{dVol}_{\mathcal{Z}}$. Suppose further that $\mathcal{Y} \times \mathcal{Z} = G(U)$, where $U$ is an open set of $\mathbb{R}^m$ and $G : U \to \mathcal{Y} \times \mathcal{Z}$ is continuously differentiable homeomorphism. If $(y, z)$ are random variables having density function $\pi_{\mathcal{Y}}(y)\pi_{\mathcal{Z}}(\cdot|y)$ with respect to the product element $\mathrm{dVol}_{\mathcal{Y}} \times \mathrm{dVol}_{\mathcal{Z}}$, then the random variable $x = G^{-1}(y, z)$ has a density with respect to Lebesgue measure if $\pi_{\mathcal{Y}}(y) \cdot \pi_{\mathcal{Z}}(z|y)$ is measurable with respect to $\mathfrak{B}(\mathcal{Y} \times \mathcal{Z}) = \mathfrak{B}(\mathcal{Y}) \times \mathfrak{B}(\mathcal{Z})$.

*Proof.* This follows as a result of proposition 8 when $\mathcal{M} = \mathcal{Y} \times \mathcal{Z}$ and $\mu_{\mathcal{M}} = \mu_{\mathcal{Y} \times \mathcal{Z}}$ is defined by

$$\mu_{\mathcal{Y} \times \mathcal{Z}}(A) = \int_A \pi_{\mathcal{Y}}(y) \cdot \pi_{\mathcal{Z}}(z|y) \, \mathrm{dVol}_{\mathcal{Z}}(z) \, \mathrm{dVol}_{\mathcal{Y}}(y) \tag{103}$$

where $A \in \mathfrak{B}(\mathcal{Y} \times \mathcal{Z})$. The conditions of the proposition are met because $U$ is an open subset of $\mathbb{R}^m$ by assumption and $G$ is a continuously differentiable homeomorphism by assumption. $\square$

The importance of the open set $U$ is best seen through an example.

**Example 1.** Let $U = \mathbb{R}^3 \setminus \{0\}$ and consider the transformation $G : U \to \mathbb{S}^2 \times \mathbb{R}_+$ by defined by

$$G(x) = (x/r, r). \tag{104}$$

where $r = \|x\|$. Thus, we see that $G$ is the spherical coordinate representation of the point $x \in U$. The inverse of $G$ is given by $G^{-1}(s, r) = rs$. Notice that by choosing $U$ to exclude the zero vector, we ensure that $G$ is indeed a homeomorphism; if the zero vector had not been included we would find that $G^{-1}(s, 0) = 0$ for all $s \in \mathbb{S}^2$ so that the inverse is not unique.

To see that $G$ is continuously differentiable, we compute

$$\frac{\partial r}{\partial x_j} = \frac{x_j}{r} \tag{105}$$

$$\frac{\partial}{\partial x_j}\left(\frac{x_i}{r}\right) = \frac{\delta_{ij}}{r} - \frac{x_i x_j}{r^3} \tag{106}$$

both of which are continuous for $x \in U$. Therefore, if $(r, s)$ is a random variable on $\mathbb{S}^2 \times \mathbb{R}_+$ with Borel-measurable density, then the random variable $G^{-1}(r, s)$ has a density in $U$ with respect to Lebesgue measure by proposition 9.

The existence of a Lebesgue measurable density in Euclidean space allows us to apply the theory of universal approximations of certain normalizing flows to guarantee that the dequantized distribution can be approximated arbitrarily well by a sufficiently expressive normalizing flow in Euclidean space.

The remaining discussion in this appendix is paraphrased from Section 3.4.3 in (Kobyzev et al., 2020), which itself draws from (Jaini et al., 2019) and (Huang et al., 2018). The essential idea is that one can produce auto-regressive normalizing flows whose couplings are dense in a space of functions that are guaranteed to have a universality property.

**Definition 7.** Let $T : \mathbb{R}^m \to \mathbb{R}^m$ be a function and write $T(x) = (T_1(x), \ldots, T_m(x))$ where $T_k : \mathbb{R}^m \to \mathbb{R}$. Such a function is called triangular if $T_k(x)$ depends only on $(x_1, \ldots, x_k)$

**Definition 8.** Let $T : \mathbb{R}^m \to \mathbb{R}^m$ be a function and write $T(x) = (T_1(x), \ldots, T_m(x))$ where $T_k : \mathbb{R}^m \to \mathbb{R}$. Such a function is called increasing if $T_k$ is an increasing function of $x_k$.

**Definition 9.** An auto-regressive normalizing flow is a triangular map $(x_1, \ldots, x_m) \mapsto (y_1, \ldots, y_m)$ of the form

$$y_k = h(x_k | \Theta_k(x_1, \ldots, x_{k-1})) \tag{107}$$

where $h(\cdot | \theta) : \mathbb{R} \to \mathbb{R}$ is a bijection parameterized by $\theta \in \mathbb{R}^d$ and $\Theta_k : \mathbb{R}^{k-1} \to \mathbb{R}^d$ is a map that parameterizes $h$ according to $(x_1, \ldots, x_{k-1})$.

In our experimentation on dequantization, we have *not* considered using auto-regressive flows in the ambient Euclidean space; neither RealNVP nor neural ODEs are auto-regressive in the sense of definition 9. Therefore, this discussion is primarily of theoretical interest.

**Definition 10.** Let $\mu$ be a probability measure on a measurable space $\mathcal{X}$ and let $x$ be a random variable taking values in $\mathcal{X}$. We say that the law of $x$ is $\mu$ if for all Borel sets $A \in \mathfrak{B}(\mathcal{X})$ we have

$$\Pr(x \in A) = \int_A \mathrm{d}\mu(x). \tag{108}$$

The following result is from (Bogachev et al., 2007).

**Proposition 10.** Let $\mu$ and $\mu'$ be probability measures that are absolutely continuous with respect to the Lebesgue measure. Let $x$ be a random variable whose law is $\mu$. Then there exists a triangular-increasing function $T$ such that the law of $T(x)$ is $\mu'$.

A construction of such a map in proposition 10 is given by the Knothe-Rosenblatt rearrangement (Villani, 2008).

The following result is from (Huang et al., 2018).

**Lemma 4.** Let $\mu$ be a probability measure that is absolutely continuous with respect to Lebesgue measure. Let $T_n$ be a sequence of measurable maps converging pointwise to a map $T$. If the law of a random variable $x$ is $\mu$, then the random variables $T_n(x)$ converge in law to to the random variable $T(x)$.

The universality of auto-regressive normalizing flows can therefore be established by demonstrating that the functions $h$ appearing in definition 9 are dense in the set of increasing monotone functions. One then applies proposition 10 and lemma 4 to demonstrate that any random variable $x$ with law $\mu$ can be transformed by an auto-regressive flow using the function $h$ into an approximation of a random variable $x'$ with law $\mu'$; the approximation has arbitrarily high fidelity as measured by convergence in law. See (Huang et al., 2018; Jaini et al., 2019) for a discussion of parameterized families of functions $h$ that are known to possess this property.

In conclusion, proposition 9 gives the necessary conditions on product manifold densities and the change-of-variables $G$ to ensure the existence of a Lebesgue measurable density in an open subset of Euclidean space. We saw in example 1 how the choice of open set allows us to avoid pathological points in Euclidean space. Using the existence of the Lebesgue-measurable density, we may apply the theory of universal approximation of Lebesgue-measurable densities using auto-regressive normalizing flows to guarantee that the dequantized distribution can be approximated arbitrarily well. We note, however, that our experiments have not used auto-regressive normalizing flows.

## M   Theory of Dequantization of Embedded Manifolds

**Definition 11.** Let $M$ be an embedded $m$-dimensional sub-manifold of $\mathbb{R}^n$. Let $U$ be an open subset of $\mathbb{R}^n$ and let $S$ be a $(n-m)$-dimensional manifold. We say that $M \times S$ can be dequantized into $U$ if there is a diffeomorphism from $(M \times S) \setminus N_{M \times S}$ into $U \setminus N_U$ where $N_{M \times S}$ and $N_U$ are negligible sets inside $M \times S$ and $U$, respectively.

**Definition 12** (Lee (2003) (page 106))**.** A regular level set is a level set consisting entirely of points $x \in \mathbb{R}^n$ such that $\nabla \Phi(x) \in \mathbb{R}^{(n-m) \times n}$ has full-rank.

**Theorem 3.** Let $M$ be a closed embedded $m$-dimensional sub-manifold of $\mathbb{R}^n$ and suppose that there exists an open neighborhood $U$ of $M$ in $\mathbb{R}^n$ and a smooth function $\Phi : U \to \mathbb{R}^{n-m}$ such that $M$ is a regular level set of $\Phi$. Then $M \times \mathbb{R}^{n-m}$ can be dequantized into an open subset of $\mathbb{R}^n$.

We will now set about proving this theorem. In all of the examples considered in the main paper $U = \mathbb{R}^n$ and later we will show that $U$ can always be taken to be the whole of $\mathbb{R}^n$.

**Definition 13** (Lee (2003) (page 138))**.** Let $M$ be a properly embedded $m$-dimensional sub-manifold of $\mathbb{R}^n$. Let $\mathrm{T}M$ be the tangent bundle of $M$ with $\mathrm{T}_x M$, the tangent space at $x$, an embedded $m$-dimensional vector space within $\mathbb{R}^n$. Denote the orthogonal complement of $\mathrm{T}_x M$ within $\mathbb{R}^n$ by $\mathrm{N}_x M$, which is an $(n-m)$-dimensional vector space. The normal bundle is

$$\mathrm{N}M = \{(x,v) \in \mathbb{R}^n \times \mathbb{R}^n : x \in M \text{ and } v \in \mathrm{N}_x M\}. \tag{109}$$

**Definition 14** (Lee (2003) (page 139))**.** Let $M$ be an embedded sub-manifold of $\mathbb{R}^n$. A tubular neighborhood of $M$ is a neighborhood $U$ of $M$ in $\mathbb{R}^n$ that is the diffeomorphic image under addition of an open subset $V \subset \mathrm{N}M$ of the form,

$$V = \{(x,v) \in \mathrm{N}M : \|v\| < \delta(x)\} \tag{110}$$

where $\delta : M \to \mathbb{R}_+$ is a continuous function.

**Lemma 5.** Let $V$ be as in eq. (110). There exists an open subset $V' \subset V$ of the form,

$$V' = \{(x,v) \in \mathrm{N}M : \|v\| < \delta'(x)\} \tag{111}$$

where $\delta' : M \to \mathbb{R}_+$ is a smooth function.

*Proof.* Consider the function $\tilde{\delta}(x) = \delta(x)/2$, which is also a continuous function. Define the continuous "error function" $\epsilon(x) = \delta(x)/4$. By the Whitney Approximation Theorem, there is a smooth function $\delta'$ that is within $\epsilon$ of $\tilde{\delta}$. Moverover, because $\delta$ is positive, $\delta'(x)$ is at least $\delta(x)/4 > 0$, hence $\delta'$ is also positive. $\square$

**Definition 15** (Lee (2003) (page 250))**.** The normal bundle is said to be smoothly trivial if it is diffeomorphic to $M \times \mathbb{R}^{n-m}$.

**Proposition 11** (Lee (2003) (page 271))**.** Let $M$ be a properly embedded $m$-dimensional sub-manifold of $\mathbb{R}^n$ and suppose that there exists an open neighborhood $U$ of $S$ in $\mathbb{R}^n$ and a smooth function $\Phi : U \mapsto \mathbb{R}^{n-m}$ such that $M$ is a regular level set of $\Phi$. Then $M$ has smoothly trivial normal bundle.

*Proof.* Given $x \in M$, it suffices to construct a smooth basis of $\mathrm{N}_x M$. By assumption $\nabla \Phi(x)$ has full-rank and therefore possesses $n-m$ independent rows. Without loss of generality, we may assume that $M$ is the zero level-set of $\Phi$; i.e. $M = \Phi^{-1}(\{0\})$ or $M = \{x \in \mathbb{R}^n : \Phi(x) = 0\}$. By differentiating the constraint, we see that all tangent vectors satisfy $v \in \mathrm{T}_x M \iff \nabla \Phi(x) v = 0$, which immediately yields that the rows of $\nabla \Phi(x)$ are orthogonal to the tangent space. Therefore, they are a basis of the normal space $\mathrm{N}_x M$. Smoothness of this basis then follows from the assumed smoothness of $\Phi$. $\square$

**Lemma 6.** The subset pre-image of the tubular neighborhood in eq. (111) is diffeomorphic to $\mathrm{N}M$.

The following proof strategy is adapted from the proof of Proposition 2.8 from Usher (2011).

*Proof.* Let $g : [0, \infty) \to [0, 1)$ be a continuous function satisfying the following properties (i) there is $\epsilon > 0$ such that $g(s) = 1$ when $s < \epsilon$, (ii) $g(s) > 0$ for every $s \in [0, \infty)$, and (iii) $\int_0^\infty g(s) \, \mathrm{d}s = 1$. We define the following function,

$$f(t) = \int_0^t g(s) \, \mathrm{d}s. \tag{112}$$

It follows from the inverse function theorem that $f$ has a continuously differentiable inverse. Now consider the map,

$$\mathrm{N}M \ni (x, v) \mapsto \left( x, \begin{cases} \delta'(x) f(\|v\|) \frac{v}{\|v\|} & \text{if } \|v\| \neq 0 \\ 0 & \text{otherwise} \end{cases} \right) \in V'. \tag{113}$$

where $\delta'$ is the smooth function defined in lemma 5. This is a smooth map because it is nothing but scalar multiplication by $\delta'(x)$ on a neighborhood of $\{(x, 0) : x \in M\}$ (when $\|v\| < \epsilon$) and otherwise is the multiplication of several smooth functions. In either case, the smooth inverse transformation is seen to be,

$$V' \ni (x, v') \mapsto \left( x, \begin{cases} f^{-1}\left( \frac{\|v'\|}{\delta'(x)} \right) \frac{v'}{\|v'\|} & \text{if } \|v'\| \neq 0 \\ 0 & \text{otherwise} \end{cases} \right) \in \mathrm{N}M. \tag{114}$$

$\square$

*Proof of Theorem 3.* Let $V$ be as in definition 14, the subset of $\mathrm{N}M$ that is diffeomorphic under addition to a tubular neighborhood of $M$. Let $V'$ be as in lemma 5, which is diffeomorphic to an open neighborhood of $M$ under addition. By lemma 6, $V'$ is itself diffeomorphic to $\mathrm{N}M$. By proposition 11, $\mathrm{N}M$ is diffeomorphic to $M \times \mathbb{R}^{n-m}$. Therefore, by chaining these diffeomorphisms together, we obtain a diffeomorphism from $M \times \mathbb{R}^{n-m}$ to an open neighborhood of $M$ in $\mathbb{R}^n$. $\square$

Now we will show that the open set into which $M \times \mathbb{R}^{n-m}$ can be dequantized in theorem 3 is almost diffeomorphic to $\mathbb{R}^n$.

**Theorem 4.** Every open subset $U$ of $\mathbb{R}^n$ is almost diffeomorphic to $\mathbb{R}^n$; that is, there exists Lebesgue-negligible sets $N_U$ and $N_{\mathbb{R}^n}$ and a diffeomorphism $\psi : U \setminus N_U \to \mathbb{R}^n \setminus N_{\mathbb{R}^n}$.

We require the following result.

**Theorem 5** (Stein & Shakarchi (2005) (page 7))**.** Every open set $U$ of $\mathbb{R}^n$ can be written as a countable union of almost disjoint closed cubes.

**Corollary 5.** Every non-empty open set $U$ of $\mathbb{R}^n$ can be written as a countably infinite union of almost disjoint closed cubes.

*Proof.* If the construction in theorem 5 is already countably infinite then there is nothing to prove. Therefore assume that $U$ is a finite union of, say, $k$ almost disjoint closed cubes. Take the $k^{\text{th}}$ cube and sub-divide it into $2^n$ sub-cubes by splitting it at the midpoint of each side dimension. This leaves us with $k - 1 + 2^n > k$ cubes and $U$ is the finite union of $k - 1 + 2^n$ almost disjoint closed cubes. This process is then iterated *ad infinitum*, thereby producing a countably infinite set of almost disjoint closed cubes whose union is $U$. $\square$

*Proof of Theorem 4.* We will use the fact that $\mathbb{Z}^n$ is countable. We will denote the interior of a set $O \subset \mathbb{R}^n$ by $\mathrm{Int}(O)$. Consider placing at each $x \in \mathbb{Z}^n$ a closed unit cube, $c_x$, whose center is $x$; the collection of these cubes is countable because $\mathbb{Z}^n$ is. Then $\mathbb{R}^n = \bigcup_{x \in \mathbb{Z}^n} c_x$ and $\mathbb{R}^n \setminus \bigcup_{x \in \mathbb{Z}^n} \mathrm{Int}(c_x)$ is a set with Lebesgue measure zero. Let $\mathcal{Q}$ denote the countably infinite set of almost disjoint closed cubes whose union is $U$. Since $\mathcal{Q}$ is countable, let $Q_k$ denote the $k^{\text{th}}$ cube; we have $U = \bigcup_{k \in \mathbb{N}} Q_k$ and $U \setminus \bigcup_{k \in \mathbb{N}} \mathrm{Int}(Q_k)$ is Lebesgue negligible. Since $\mathbb{Z}^n$ is countable, there exists a bijection $\omega : \mathbb{N} \to \mathbb{Z}^n$. By shifting and scaling $\mathrm{Int}(Q_k)$ (which

are, of course, smooth and invertible operations) we can transform it into $\text{Int}(c_{\omega(k)})$. In particular, suppose $Q_k$ is a closed cube of side-length $\ell_k \in \mathbb{R}_+$ centered at $q^{(k)} \in \mathbb{R}^n$. Then we write,

$$\text{Int}(Q_k) = \left\{ x \in \mathbb{R}^n : \max_{i \in \{1,\ldots,n\}} \left| x_i - q_i^{(k)} \right| < \ell_k \right\}. \tag{115}$$

By scaling $\text{Int}(Q_k)$ be $1/\ell_k$ we obtain,

$$\frac{1}{\ell_k} \text{Int}(Q_k) = \left\{ x \in \mathbb{R}^n : \max_{i \in \{1,\ldots,n\}} \left| x_i - \frac{q_i^{(k)}}{\ell_k} \right| < 1 \right\}, \tag{116}$$

which is the cube of unit side-length whose center is $q^{(k)}/\ell_k$. Now shift this cube by $\omega(k) - q^{(k)}/\ell_k$ to produce $\text{Int}(c_{\omega(k)})$. Applying this transformation for every $k$ allows us to construct the map,

$$\psi(y) = \begin{cases} \frac{y}{\ell_1} + \omega(1) - \frac{q^{(1)}}{\ell_1} & \text{if } y \in \text{Int}(Q_1) \\ \frac{y}{\ell_2} + \omega(2) - \frac{q^{(2)}}{\ell_2} & \text{if } y \in \text{Int}(Q_2) \\ \quad \vdots \end{cases} \tag{117}$$

and this function is a diffeomorphism from $\bigcup_{k \in \mathbb{N}} \text{Int}(Q_k)$ to $\bigcup_{x \in \mathbb{Z}^n} \text{Int}(c_x)$, which are equivalent to $U$ and $\mathbb{R}^n$, respectively, up to sets of Lebesgue measure zero. $\qquad \square$

Theorem 4 allows us to give the following modification of theorem 3.

**Corollary 6.** Let $M$ be a closed embedded $m$-dimensional sub-manifold of $\mathbb{R}^n$ and suppose that there exists an open neighborhood $U$ of $M$ in $\mathbb{R}^n$ and a smooth function $\Phi : U \to \mathbb{R}^{n-m}$ such that $M$ is a regular level set of $\Phi$. Then $M \times \mathbb{R}^{n-m}$ can be dequantized into $\mathbb{R}^n$.

*Proof.* By theorem 3, there is an open subset of $\mathbb{R}^n$ into which $M \times \mathbb{R}^{n-m}$ may be dequantized. Apply theorem 4 to this open subset. $\qquad \square$

# N Derivation of the Jacobian Determinant for Hyperspherical Coordinates

**Definition 16** (Hyperspherical Coordinates). Let $x = (x_1, \ldots, x_n)$ be an element of $\mathbb{R}^n$ such that $x \neq 0$. The hyperspherical coordinates of $x$ are $(y, r) \in \mathbb{S}^{n-1} \times \mathbb{R}_+$ where

$$r \stackrel{\text{def.}}{=} \sqrt{x_1^2 + \ldots + x_n^2} \tag{118}$$

$$y \stackrel{\text{def.}}{=} (x_1/r, \ldots, x_n/r) \tag{119}$$

Via the natural inclusion of $\mathbb{S}^{n-1}$ into $\mathbb{R}^n$, hyperspherical coordinates may be viewed as a subset of $\mathbb{R}^{n+1}$.

**Proposition 12.** As a mapping from $\mathbb{R}^n$ to $\mathbb{S}^{n-1} \times \mathbb{R}_+$, the transformation $x \mapsto (y, r)$ has Jacobian determinant $1/r^{n-1}$.

To prove this result, we require the matrix determinant lemma as follows.

**Lemma 7** (Matrix Determinant Lemma). Let $A$ be an invertible $n \times n$ matrix and let $u, v \in \mathbb{R}^n$. Then

$$\det(A + uv^\top) = (1 + u^\top A^{-1} v) \cdot \det(A). \tag{120}$$

Our proof now proceeds by computing the Jacobian of the transformation in the ambient space and apply theorem 1.

**Lemma 8** (Partial Derivatives).

$$\frac{\partial r}{\partial x_j} = \frac{x_j}{r} \tag{121}$$

$$\frac{\partial}{\partial x_j}\left(\frac{x_i}{r}\right) = \frac{\delta_{ij}}{r} - \frac{x_i x_j}{r^3} \tag{122}$$

*Proof.* For the first equality:

$$\frac{\partial}{\partial x_j} r = \frac{1}{2r} 2x_j = \frac{x_j}{r} \tag{123}$$

For the second equality:

$$\frac{\partial}{\partial x_j}\left(\frac{x_i}{r}\right) = \frac{\left(\frac{\partial}{\partial x_j} x_i\right) r - x_i \left(\frac{x_j}{r}\right)}{r^2} \tag{124}$$

$$= \frac{\delta_{ij}}{r} - \frac{x_i x_j}{r^3} \tag{125}$$

$\square$

**Corollary 7.** The Jacobian of the coordinate transformation may be expressed as,

$$J = \begin{pmatrix} A \\ b^\top \end{pmatrix} \tag{126}$$

$$A \stackrel{\text{def.}}{=} \left(\frac{1}{r}\text{Id} - \frac{xx^\top}{r^3}\right) \tag{127}$$

$$b^\top \stackrel{\text{def.}}{=} \begin{pmatrix} \frac{x_1}{r} & \cdots & \frac{x_n}{r} \end{pmatrix} \tag{128}$$

*Proof.* By the preceding corrolary and observing that $A$ is a symmetric matrix, we have

$$J^\top J = AA + bb^\top. \tag{129}$$

Expanding $AA$ gives:

$$AA = \left(\tfrac{1}{r}\text{Id} - \tfrac{xx^\top}{r^3}\right)\left(\tfrac{1}{r}\text{Id} - \tfrac{xx^\top}{r^3}\right) \tag{130}$$

$$= \frac{\text{Id}}{r^2} - \frac{xx^\top x^\top x}{r^6} \tag{131}$$

$$= \frac{\text{Id}}{r^2} - \frac{xx^\top}{r^4} \tag{132}$$

since $x^\top x = r^2$. Moreover $bb^\top = \frac{xx^\top}{r^2}$. Therefore,

$$J^\top J = \frac{1}{r^2}\text{Id} - \frac{xx^\top}{r^4} + \frac{xx^\top}{r^2} \tag{133}$$

$$= \frac{1}{r^2}\text{Id} + \left(\frac{1}{r^2} - \frac{1}{r^4}\right)xx^\top \tag{134}$$

Now applying the matrix determinant lemma, we have,

$$\det(J^\top J) = \left(1 + r^4\left(\frac{1}{r^2} - \frac{1}{r^4}\right)\right)\frac{1}{r^{2n}} \tag{135}$$

$$= \frac{r^2}{r^{2n}} \tag{136}$$

$$= \frac{1}{r^{2n-2}}. \tag{137}$$

Taking the square-root completes the proof. □

