# OpenReview forum: "Manifold Density Estimation via Generalized Dequantization"
_TMLR — Rejected by TMLR_

### Review · Reviewer_AX3k · 2023-03-20

**Summary Of Contributions:**

This paper considers the important problem of density estimation on manifolds. They do this via a technique they call dequantization. Specifically, given a Manifold M, they find a space S such that M x S is isomorphic to R^n for some n. The paper then provides a way to characterize distributions on M using distributions on R^n and S. They then parameterize these distributions and provide a tractable lower bounded for log-likelihood and approximate the maximum likelihood estimator. Experimentally they show that their technique works well.

**Audience:**

Yes

**Claims And Evidence:**

Yes

**Requested Changes:**

Please add details for the parameterization, the training details, model sizes, etc, as well as a discussion on importance sampling.

**Strengths And Weaknesses:**

In terms of strength, the idea in the paper is very simple and easy to implement. It is flexible and adaptable to a variety of situations and has decent theoretical backing. However, the parameterization they need is using a normalizing flow type method and hence need significantly more dimensions to parameterize the density than the dimension of the manifold, but this is minor for me.

Further. I think the paper would benefit from adding some details. Some more details on the parameterization would be nice. Further, the authors do not really discuss importance sampling and more details on this would be appreciated. Finally, I think a discussion of the connection to ELBO and how tight of an approximation the lower bound is needed.

Could you also clarify what you mean by isomorphic? Are these isomorphisms of vector spaces? What is the vector space structure on $\mathbb{S}^1 \times \mathbb{R}$?

---

> ### Author Response · Authors · 2023-05-17
> **Answer to AX3k**
>
> Thank you for your comments. We are pleased that you found the idea simple and easy to implement.
> We are uploading a revision based on the reviews, with changes in red.
>
>
> Specific comments:
> * isomorphic - clarified.
>
> * Details. The code provided with the paper is publicly available on github (link omitted due to anonymity requirements). We included with the code the parameters used in all the experiments in the paper. If the reviewer finds that this is insufficient, we will add additional information to the final version of the paper.
>
> * Importance sampling - the auxiliary distribution/ importance sampling distribution is used to evaluate the integral/ expectation over the nuance variable. Is the reviewer requesting additional discussion of the use of this auxiliary distribution in general, or additional discussion of the IS variant of the algorithm?

---

### Review · Reviewer_rgh9 · 2023-03-31

**Summary Of Contributions:**

This paper proposes a method to approximate the density when the data lie on predefined manifolds. The idea is to see the manifold as the quantized-space and the data can be dequantized by moving them stochastically in the ambient space in a certain way depending on the actual manifold. Then a normalizing flow is used to model the dequantized distribution in the ambient space, which can then be quantized (mapped back) to the manifold. The effectiveness of the method is demonstrated in synthetic experiments and predefined manifolds.

**Audience:**

Yes

**Broader Impact Concerns:**

As regards the current paper, there are no direct ethical implications. Of course, as it falls within the generative models domain, indirect implications might occur, but the possibility for this work is low. The authors claim that this work might have a societal impact as a tool to be used in other disciplines. x

**Claims And Evidence:**

Yes

**Requested Changes:**

These questions/clarifications are mainly for a better understanding of the paper:
- In Eq. 1 (and the sphere-manifold) I would like to see the computation of the Jacobian determinant. I think for the rest of the cases this is beneficial information to have as well.
- What is the auxiliary manifold that you mention in the paper. I suppose that this is the dequantized manifold?
- Why the lower bound (ELBO) Eq. 4 is necessary to train the method?
- As regards the distribution of the dequantization density (e.g. in Eq. 4), I think this should have particular properties to enable backpropagation and training the model. For example, the re-parametrization trick should be feasible? Perhaps a different approach is used to train this distribution?
- Can we see graphically in some examples the distribution that the normalizing flow learns? For example, for the density on a circle the dequantized distribution lies in R^2.
- I think that the related work that is missing should be included.
- I believe that additional experiments with real-world (directional) data should be included (perhaps for the Stiefel manifold as well).
- For the sphere example, maybe some additional examples can be included in the appendix.
- I think that it is hard to understand from the tables what constitutes a good behavior i.e. are higher/lower values better?


**Strengths And Weaknesses:**

Strengths:
- The idea is interesting as it provides a modular way to learn densities on manifolds. In principle, one part is the quantization-dequantization process and then is the normalizing flow part modeling part.
- Also, compared to related works this approach is in some sense simpler and easier to use.
- The paper is ok written, but I think that it can be improved in some cases.

Weaknesses:
- The method relies on pre-specified manifolds and quantization/dequantization techniques. It would be nice if there is an approach for general manifolds.
- The experimental section is a bit hard to interpret/understand and perhaps more experiments are necessary to verify its usefulness.
- Some figures can help the understanding of the methodology.
- There is a related work [1] that is not cited but it is very similar to the proposed approach. I think that this method should be discussed in the manuscript and perhaps included in the experimental section.

[1] "Density estimation on low-dimensional manifolds: an inflation-deflation approach", Christian Horvat, Jean-Pascal Pfister, arxiv, 2021.

---

> ### Author Response · Authors · 2023-05-17
> **Answer to rgh9**
>
> Thank you for your comments. We are pleased that you found the method interesting and simpler and easier to implement that related work. We note that the method can be applied to other manifolds, but each new manifold indeed requires a derivation of a new decomposition.
> We are uploading a revision based on the reviews, with changes in red.
>
> Specific comments:
> * “Eq. 1…” - Added.
> * the auxiliary manifold - added clarification.
> * ELBO Eq. 4 - The ELBO is used to train the model because computing the exact marginal distribution at a point on the manifold would require computing the preimage of those points in the ambient Euclidean space that get mapped to that point on the manifold. Integrating over this set seems hard which is why we use the variational lower bound and importance sampling instead. In contrast, ELBO admits conveniently to stochastic gradient descent and circumvents the need for exact integration in the context of a stochastic gradient descent algorithm.
> * particular properties of the distribution of the dequantization. - The dequantization density indeed needs to be one that allows automatic differentiation. However, we note that ``most of the work’’ goes into estimating the distribution in the ambient space using the standard normalizing flows where the reparametrization trick is employed. We note that the dequantization distribution itself can even be a fixed distribution.
> * I think that it is hard to understand from the tables what constitutes a good behavior i.e. are higher/lower values better? - Added clarification. - Added
> * Citations - citation added.
> * We apologize that due to an error in planning on our part we have not been able to add additional examples. The examples we included in the paper are motivated by examples found in the cited literature.

---

### Review · Reviewer_T9wV · 2023-05-03

**Summary Of Contributions:**

The paper proposes a reasonably general mechanism for estimating densities on submanifolds of $\mathbb{R}^n$. This works by building a distribution in the ambient space and then marginalizing the dimensions that are 'orthogonal' to the manifold. This marginalization is carried out with importance sampling (IS). Since IS can be problematic during learning (slow, unstable), the paper also provides a simple lower bound to the likelihood of the manifold density, which is suitable for optimization.

**Audience:**

Yes

**Claims And Evidence:**

Yes

**Requested Changes:**

* I think the authors should cite the projected normal distribution as it is essentially the motivating example. The "Directional Statistics" textbook by Mardia & Jupp covers this distribution.
* I think the statement (Sec 3): "In contrast to these approaches, the method proposed here allows for the use of any density estimation technique defined on the ambient Euclidean space." should be complemented with a remark about the requirements that you can figure out how to decompose the distribution in ambient space in order to apply the method (while feasible for the presented manifolds, it may not be universally trivial).
* As a general comment, it would be nice to have proper usage of the  \citet and \citep LaTeX commands, e.g. "Originally introduced by (Uria et al., 2013)" should be "Originally introduced by Uria et al. (2013)". That is, the example should be using \citet.
* Section 4.2.1: I think $\mathbb{S}^m$ should be $\mathbb{S}^{m-1}$ for the paper to be self-consistent regarding notation.
* In theorem 1, I assume that the invertibility places constraints on dimensions of $\mathcal{Y}$ and $\mathcal{Z}$, e.g. $m = dim(\mathcal{Y})+dim(\mathcal{Z})$. If that is correct, it would be good to state it explicitly.
* Section 5.1.3: I miss a citation for the claim "I assume that invertebility places constraints on dimensions of Y and Z, e.g. m = dim(Z)+dim(Y). Is that correct, if so, it would be good to state explicitly."

**Strengths And Weaknesses:**

Strengths:
* The formulation is simple and easy to understand while solving a real problem (density estimation on manifolds).
* The paper is well-organized and reads well; in particular, the motivating example is very helpful.
* Empirical results are informative.

Weaknesses:
* The empirical results aren't overwhelming, but they need not be.
* I have a couple of suggestions for changes (below), but only minor stuff.

---

> ### Author Response · Authors · 2023-05-14
> **Missing citation**
>
> In your last bullet point, you have asked for a citation for a particular claim. However, the quoted text is your immediately preceding request. What is the quoted text you meant to ask about?

---

> ### Author Response · Authors · 2023-05-17
> **Answer to T9wV**
>
> Thank you for your comments. We are pleased that you found that the paper is easy to understand, that it solves a real problem, and that the results are informative.
> We are uploading a revision based on the reviews, with changes in red.
>
> Regarding specific comments:
> * projected normal distribution - added citation
> * “In contrast to these approaches…” - added.
> * Citation style - updated.
> * Section 4.2.1. And notation - corrected, thank you.
> * Invertibility - clarified.

---

### Decision · Action_Editors · 2023-07-14

**Recommendation:** Reject

**Comment:**

- I think Fig. 1 is not self-contained. Reading it without looking at Section 2 is quite mysterious, and I am afraid casual readers might be discouraged upon not understanding what should be the "key easy example" of the paper. I suggest to move this Figure to Section 2, and/or think about a different, simpler example (maybe 2d?)

- Fig. 2 is supposed to provide a "roadmap" but is difficult to understand as it is. First, the arrows are circular, without this being addressed in the test. It references immediately Eq. 6, but I do not understand Eq. 6, or I am missing notations. You write det(∇G(x)^T ∇G(x)) , yet G is not a real-valued map IIUC. What does the nabla symbol stand for in that case, if not a gradient operator? https://en.wikipedia.org/wiki/Del
I can see this is clarified in the appendix between 61 and 62 but it's annoying to have to peek into the appendix to start grasping the "leading" figure of the paper.

- You mention "see inter alia Rezende et al. (2020)" to introduce Eq.6, but I have failed to find an exact or approximate match. They do use Jacobians in Eq. 20, which sounds more intuitive

- While I appreciate this method is not immediately sold as being practical, I would expect a minimal amount of discussion on the stability of the training procedure, stepsize, etc, notably when compared with other competing methods. What was the hyperparameter tuning effort for these methods, compared to those used as baselines? The gap in performance is fairly small, and somewhat disappointing. It would be reassuring to think that this was not obtained by over parameterizing the new proposals and not paying attention to baselines.

- A previous version of this work appeared in a workshop. --> please provide more details

- In equation 10, it might help the reader to do away with the notation x =... and directly write the ratio as a function of y,z only, to be consistent with Eq. 13 or 14.



**Audience:**

This is a fairly interesting topic that is of interest to the more "geometric" inclined sections of the ML readership. While the findings and methods are overall relevant, the short and somewhat limited experimental evaluation can be improved.

**Claims And Evidence:**

The authors propose a method to perform density estimation on manifolds Y from data. Essentially the method relies on finding a mapping from an ambient Euclidean space (in which one can easily parameterize densities) that can be mapped bijectively to a product Y x Z where Z is an auxiliary space of variables, i.e. exists invertible G X -> Y x Z.

The authors propose then to approximate a density on Y by push-forwarding the learned density on the Euclidean space, and marginalizing on Z, i.e. P(y) = π_y (G # P_X) where π_y is the marginalization operator. This marginalization is approximated by proposing an explicit "prior" density on Z, i.e. using important sampling, see Eq. 10.

The reviewers are leaning towards accept. I will play the devil's advocate, because I believe some of the points raised by some of them (notably w.r.t. clarity) have not been completely addressed. The paper clearly needs one more iteration before being publishable. I encourage the authors to resubmit taking into account these aspects to improve readability, and, ultimately, the impact of their work.

**Resubmission Of Major Revision:**

The authors may consider submitting a major revision at a later time.